# On the Robustness of Transformers against Context Hijacking for Linear Classification

## Abstract

Transformer-based Large Language Models (LLMs) have demonstrated powerful in-context learning capabilities. However, their predictions can be disrupted by factually correct context, a phenomenon known as context hijacking, revealing a significant robustness issue. To understand this phenomenon theoretically, we explore an in-context linear classification problem based on recent advances in linear transformers. In our setup, context tokens are designed as factually correct query-answer pairs, where the queries are similar to the final query but have opposite labels. Then, we develop a general theoretical analysis on the robustness of the linear transformers, which is formulated as a function of the model depth, training context lengths, and number of hijacking context tokens. A key finding is that a well-trained deeper transformer can achieve higher robustness, which aligns with empirical observations. We show that this improvement arises because deeper layers enable more fine-grained optimization steps, effectively mitigating interference from context hijacking. This is also well supported by our numerical experiments. Our findings provide theoretical insights into the benefits of deeper architectures and contribute to enhancing the understanding of transformer architectures.

## 1. Introduction

Transformers (Vaswani et al., 2017) have demonstrated remarkable capabilities in various fields of deep learning, such as natural language processing (Radford et al., 2019; Achiam et al., 2023; Vig & Belinkov, 2019; Touvron et al., 2023; Ouyang et al., 2022; Devlin, 2018). A common view of the superior performance of transformers lies in its re-markable in-context learning ability (Brown, 2020; Chen et al., 2022; Liu et al., 2023a), that is, transformers can flexibly adjust predictions based on additional data given in context contained in the input sequence itself, without updating parameters. This impressive ability has triggered a series of theoretical studies attempting to understand the in-context learning mechanism of transformers (Olsson et al., 2022; Garg et al., 2022; Xie et al., 2021; Guo et al., 2023; Wu et al., 2023). These studies suggest that transformers can behave as meta learners (Chen et al., 2022), implementing certain meta algorithms (such as gradient descent (Von Oswald et al., 2023; Ahn et al., 2023; Zhang et al., 2024b) based on context examples, and then applying these algorithms to the queried input.

Despite the benefits of in-context learning abilities in transformers, this feature can also lead to certain negative impacts. Specifically, while well-designed in-context prompts can help generate desired responses, they can also mislead the transformer into producing incorrect or even harmful outputs, raising significant concerns about the robustness of transformers (Chowdhury et al., 2024; Liu et al., 2023c; Zhao et al., 2024). For instance, a significant body of work focuses on jailbreaking attacks (Chao et al., 2023; Niu et al., 2024; Shen et al., 2024; Deng et al., 2023; Yu et al., 2023), which aim to design specific context prompts that can bypass the defense mechanisms of large language models (LLMs) to produce answers to dangerous or harmful questions (e.g., "how to build a bomb?"). It has been demonstrated that, as long as the context prompt is sufficiently long and flexible to be adjusted, almost all LLMs can be successfully attacked (Anil et al., 2024). These studies can be categorized under adversarial robustness, where an attacker is allowed to perturb the contextual inputs arbitrarily to induce the transformer model to generate targeted erroneous outputs (Shi et al., 2023; Pandia & Ettinger, 2021; Creswell et al., 2022; Yoran et al., 2023).

However, in addition to the adversarial attack that may use harmful or incorrect context examples, it has been shown that the predictions of LLMs can also be disrupted by harmless and factually correct context. Such a phenomenon is referred to as *context hijacking* (Jiang et al., 2024; Jeong, 2023), which is primarily discovered on fact retrieval tasks, i.e. the output of the LLMs can

[1]Anonymous Institution, Anonymous City, Anonymous Region, Anonymous Country. Correspondence to: Anonymous Author <anon.email@domain.com>.

Preliminary work. Under review by the International Conference on Machine Learning (ICML). Do not distribute.

| Model | Context Input | Output |
|---|---|---|
| **All models** | Rafael Nadal's best sport is | tennis |
| **GPT2-SMALL** (12 Layers) | 2×{Rafael Nadal is not good at playing basketball.} Rafael Nadal's best sport is | basketball |
| **GPT2-MEDIUM** (24 Layers) | 8×{Rafael Nadal is not good at playing basketball.} Rafael Nadal's best sport is | basketball |
| **GPT2-LARGE** (36 Layers) | 11×{Rafael Nadal is not good at playing basketball.} Rafael Nadal's best sport is | basketball |

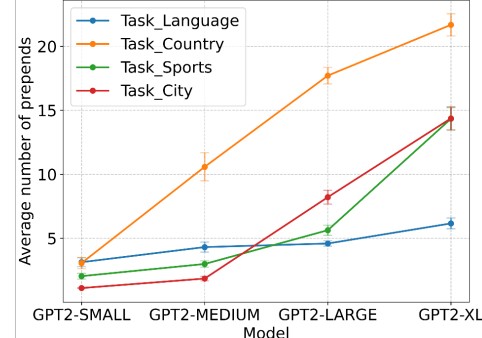

*Figure 1.* **Context hijacking phenomenon in LLMs of different depths.** *Left*: If there are no or only a few factually correct prepends, LLMs of different depths can correctly predict the next token. When the number of prepends increases, the outputs of models are disrupted, and deeper models are more robust. *Right*: Four different types of tasks are introduced, each with a fixed template, and tested on LLMs of different depths. The horizontal axis is the model with depth from small to large, and the vertical axis is the average number of prepends required to successfully interfere with the model output. Experiments show that deeper models perform more robustly. (Experimental setup is given in Appendix G.1)

be simply manipulated by modifying the context with additional factual information. For example, as shown in Figure 1, the GPT2 model can correctly answer the question "Rafael Nadal's best sport is" with "tennis" when giving context examples. However, if factually correct context examples such as "Rafael Nadal is not good at playing basketball" are provided before the question, the GPT-2 model may incorrectly respond with "basketball". Then, it is interesting to investigate whether such a phenomenon depends on different tasks and transformer architectures. To this end, we developed a class of context hijacking tasks and counted the number of context examples that led to incorrect outputs (see Figure 1). Our findings indicate that increasing the number of prepended context examples amplifies the effect on the transformer's prediction, making it more likely to generate incorrect outputs. Additionally, we observed that deeper transformer models exhibit higher robustness to context hijacking, requiring more prepended context examples to alter the model's output. Therefore, conducting a precise robustness analysis regarding context hijacking could provide valuable insights in understanding the architecture of the transformer model.

In this paper, we aim to develop a comprehensive theoretical analysis on the robustness of transformer against context hijacking. In particular, we follow the general design of many previous theoretical works (Olsson et al., 2022; Ahn et al., 2023; Frei & Vardi, 2024) on the in-context learning of transformers, by considering the multi-layer linear transformer models for linear classification tasks, where the hijacking examples are designed as the data on the boundary but with an opposite label to the queried input. Starting from the view that the $L$-th transformer models can implement $L$-step gradient descent on the context examples, with an arbitrary initialization, we formulate the transformer training

as finding the optimal multi-step gradient descent methods with respect to the learning rates and initialization. Then, we prove the optimal multi-step gradient strategy, and formulate the optimal learning rate and initialization as the function of the iteration number (i.e., model depth) and the context length. Furthermore, we deliver the theoretical analysis on the robustness based on the proved optimal gradient descent strategy, which shows that as the transformer become deeper, the corresponding more fine-grained optimization steps can be less affected by the hijacking examples, thus leading to higher robustness. This is well aligned with the empirical findings and validated by our numerical experiments. We summarize the main contributions of this paper as follows:

- We develop the first theoretical framework to study the robustness of multi-layer transformer model against context hijacking, where the hijacked context example is designed as the data with the factually correct label but close to the prediction boundary. This is different from a very recent related work on the robustness of transformer (Anwar et al., 2024) that allows the context data to be arbitrarily perturbed, which could be factually incorrect.
- Based on the developed theoretical framework, we formulate the test robust accuracy of the transformer as a function with respect to the training context length, number of hijacked context examples, and the depth of the transformer model. The key of our analysis is that we model the in-context learning mechanism of a well-trained multi-layer transformer as an optimized multi-step gradient descent, where the corresponding optimal initialization and learning rates can be precisely characterized. This could of independent interest to other problems that involve the gradient descent methods on linear problems.
- Based on the developed theoretical results, we demonstrate that deeper transformers are more robust because they are able to perform more fine-grained optimization

steps on the context samples, which can potentially explain the practical observations of LLMs in the real world (see Figure 1). The theoretical results are well supported by synthetic numerical experiments in various settings.

## 2. Related works

**In-context learning via transformers.** The powerful performance of transformers is generally believed to come from its in-context learning ability (Brown, 2020; Chen et al., 2022; Min et al., 2022; Liu et al., 2023a; Xie et al., 2021). A line of recent works study the phenomenon of in-context learning from both theoretical (Bai et al., 2024; Guo et al., 2023; Lin et al., 2023; Chen et al., 2024; Frei & Vardi, 2024; Huang et al., 2023; Siyu et al., 2024; Li et al., 2024) and empirical (Garg et al., 2022; Akyürek et al., 2022a; Li et al., 2023a; Raventós et al., 2024; Pathak et al., 2023; Panwar et al., 2023; Bhattamishra et al., 2023; Fu et al., 2023; Lee et al., 2024) perspectives on diverse settings. Brown (2020) first showed that GPT-3 can perform in-context learning. Chen et al. (2024) studied the role of different heads within transformers in performing in-context learning focusing on the sparse linear regression setting. Frei & Vardi (2024) studied the ability of one-layer linear transformers to perform in-context learning for linear classification tasks.

**Mechanism interpretability of transformers.** Among the various theoretical interpretations of transformers (Friedman et al., 2024; Yun et al., 2020; Dehghani et al., 2019; Lindner et al., 2024; Pandit & Hou, 2021; Pérez et al., 2021; Bills et al., 2023; Wei et al., 2022; Weiss et al., 2021; Zhou et al., 2023; Chen & Zou, 2024), one of the most widely studied theories is the ability of transformers to implement optimization algorithms such as gradient descent (Von Oswald et al., 2023; Ahn et al., 2023; Zhang et al., 2024b; Bai et al., 2024; Wu et al., 2023; Cheng et al., 2023; Akyürek et al., 2022b; Dai et al., 2023; Zhang et al., 2024a). Von Oswald et al. (2023) theoretically and empirically proved that transformers can learn in-context by implementing a single step of gradient descent per layer. Ahn et al. (2023) theoretically analyzed that transformers can learn to implement preconditioned gradient descent for in-context learning. Zhang et al. (2024b) considered ICL in the setting of linear regression with a non-zero mean Gaussian prior, a more general and common scenario where different tasks share a signal, which is highly relevant to our work.

**Robustness of transformers.** The security issues of large language models have always attracted a great deal of attention (Yao et al., 2024; Liu et al., 2023b; Perez & Ribeiro, 2022; Zou et al., 2023; Apruzzese et al., 2023). However, most of the research focuses on jail-breaking black-box models (Chowdhury et al., 2024), such as context-based adversarial attacks (Kumar et al., 2023; Wei et al., 2023; Xu

et al., 2023; Wang et al., 2023a; Zhu et al., 2023; Cheng et al., 2024; Wang et al., 2023b). There is very little white-box interpretation work of attacks on the transformer, the foundation model of LLMs (Qiang et al., 2023; Bailey et al., 2023; He et al., 2024; Anwar et al., 2024; Jiang et al., 2024). Qiang et al. (2023) first considered attacking large language models during in-context learning, but they did not study the role of transformers in robustness. Jiang et al. (2024) proposed the phenomenon of context hijacking, which became the key motivation of our work. They analyzed this problem from the perspective of associative memory models instead of the in-context learning ability of transformers.

## 3. Preliminaries

In this section, we will provide a detailed introduction to our setup of the context hijacking problem, including the data model, transformer architecture, and evaluation metric.

### 3.1. Data model

To understand the mechanism of context hijacking phenomenon, we model it as a binary classification task, where the query-answer pair is modeled as the input-response pair $((\mathbf{x}, y) \in \mathbb{R}^d \times \{\pm 1\})$. In particular, we present the definition of the data model as follows:

**Definition 3.1** (Data distribution). Let $\mathbf{w}^* \in \mathbb{R}^d$ be a vector drawn from a prior distribution on the $d$ dimensional unit sphere $\mathbb{S}^{d-1}$, denoted by $p_{\boldsymbol{\beta}^*}(\cdot)$, where $\boldsymbol{\beta}^* \in \mathbb{S}^{d-1}$ denotes the expected direction of $\mathbf{w}^*$. Then given the generated $\mathbf{w}^*$, the data pair $(\mathbf{x}, y)$ is generated as follows: the feature vector is $\mathbf{x} \sim \mathcal{N}(\mathbf{0}_d, \mathbf{I}_d)$ and the corresponding label is $y = \text{sign}(\langle \mathbf{w}^*, \mathbf{x} \rangle)$.

Based on the data distribution of each instance, we then introduce the detailed setup of the in-context learning task in our work. In particular, we consider the setting that the transformer is trained on the data with clean context examples and evaluated on the data with hijacked context.

**Training phase.** During the training phase, we are given $n$ clean context examples $\{(\mathbf{x}_1, y_1), \ldots, (\mathbf{x}_n, y_n)\}$ and a query $\mathbf{x}_{\text{query}}$ with its label $y_{\text{query}}$. In particular, here we mean the clean examples as the $\{(\mathbf{x}_1, y_1), \ldots, (\mathbf{x}_n, y_n)\}$ are drawn from the same data distribution $\mathcal{D}_{\mathbf{w}^*}$ as $(\mathbf{x}_{\text{query}}, y_{\text{query}})$. Then, the input data matrix for in-context learning is designed as follows:

$$\mathbf{Z} = \begin{bmatrix} \mathbf{x}_1 & \cdots & \mathbf{x}_n & \mathbf{x}_{\text{query}} \\ y_1 & \cdots & y_n & 0 \end{bmatrix} \in \mathbb{R}^{(d+1) \times (n+1)}. \quad (3.1)$$

Here, to ensure that the dimension of $\mathbf{x}_{\text{query}}$ aligns with those of other input pair $(\mathbf{x}_i, y_i)$, we concatenate it with $0$ as a placeholder for the unknown label $y_{\text{query}}$. Ideally, we anticipate that given the input $\mathbf{Z}$, the output of the transformer model, denoted by $\widehat{y}_{\text{query}}$ can match the ground truth

one. Moreover, we also emphasize that within each data matrix $\mathbf{Z}$, the context examples and the queried data should be generated based on the same ground truth vector $\mathbf{w}^*$, while for different input matrices, e.g., $\mathbf{Z}$ and $\mathbf{Z}'$, we allow their corresponding ground truth vectors could be different, which are i.i.d. drawn from the prior $p_{\boldsymbol{\beta}^*}(\cdot)$.

The training data distribution simulates the pre-training data of the large language model. Unlike existing works (Ahn et al., 2023; Olsson et al., 2022) where the prior of $\mathbf{w}^*$ is assumed to have a zero mean, we consider a setting where $\mathbf{w}^*$ has a non-zero mean (i.e., $\boldsymbol{\beta}^*$). This approach is inspired by empirical observations (see Figure 1) that transformer models can perform accurate zero-shot predictions. Consequently, our model can encapsulate both memorization and in-context learning, where the former corresponds to recovering the mean of the prior distribution, i.e., $\boldsymbol{\beta}^*$, and the latter aims to manipulate the $\{(\mathbf{x}_1, y_1), \ldots, (\mathbf{x}_n, y_n)\}$ effectively. In contrast, existing works primarily focus on the latter, thereby failing to fully explain the interplay between memorization and in-context learning.

**Test phase.** During the test phase, context examples are designed based on the query input $\mathbf{x}_{\text{query}}$ to effectively execute the attack. Inspired by empirical observations (Figure 1) and prior experience with jailbreaking attacks (Anil et al., 2024), we choose to use repeated hijacking context examples during the test phase. Specifically, since the hijacked context should be factually correct, we consider data similar to the queried input but with a correct and opposite label of low confidence. Mathematically, this involves projecting $\mathbf{x}_{\text{query}}$ onto the classification boundary. To this end, given the target query data $(\mathbf{x}_{\text{query}}, y_{\text{query}})$, we formalize the design of the hijacked context example as follows.

**Definition 3.2** (Hijacked context data). Let $(\mathbf{x}, y)$ be a input pair and $\mathbf{w}^*$ be the corresponding ground truth vector. Additionally, denote $\mathbf{x}_\perp$ as the projection of $\mathbf{x}$ on the boundary of classifier, i.e. $\mathbf{x}_\perp = (\mathbf{I}_d - \mathbf{w}^*(\mathbf{w}^*)^\top) \cdot \mathbf{x}$. Then, the query pair $(\mathbf{x}_{\text{query}}, y_{\text{query}})$ is generated as $\mathbf{x}_{\text{query}} = \mathbf{x}_\perp + \sigma \mathbf{w}^*$ and $y_{\text{query}} = \text{sign}\left(\langle \mathbf{w}^*, \mathbf{x}_{\text{query}} \rangle\right) = \text{sign}(\sigma)$ with $\sigma$ being a random variable, and the hijacked context example is designed as $\mathbf{x}_{\text{hc}} = \mathbf{x}_\perp$ and $y_{\text{hc}} = -y_{\text{query}}$.

Note that we pick $\langle \mathbf{x}_{\text{hc}}, \mathbf{w}^* \rangle = 0$ to enforce hijacked context lies on the boundary of the classifier. A more rigorous design is to set $\mathbf{x}_{\text{hc}} = \mathbf{x}_\perp - \eta \cdot y_{\text{query}} \cdot \mathbf{w}^*$ for some positive quantity $\eta$, where it can be clearly shown that $y_{\text{hc}} = \text{sign}(\langle \mathbf{x}_{\text{hc}}, \mathbf{w}^* \rangle) = -y_{\text{query}}$. Definition 3.2 concerns the limiting regime by enforcing $\eta \to 0^+$.

Then, based on the above design, the input data matrix in the test phase is constructed as follows:

$$\mathbf{Z}^{\text{hc}} = \begin{bmatrix} \mathbf{x}_{\text{hc}} & \cdots & \mathbf{x}_{\text{hc}} & \mathbf{x}_{\text{query}} \\ y_{\text{hc}} & \cdots & y_{\text{hc}} & 0 \end{bmatrix} \in \mathbb{R}^{(d+1)\times(N+1)}. \tag{3.2}$$

Here we use $N$ to denote the number of hijacked context examples. The example $(\mathbf{x}_{\text{hc}}, y_{\text{hc}})$ can also be interpreted to the closest data to $\mathbf{x}_{\text{query}}$ but with a different label $-y_{\text{query}}$, which principally has the ability to perturb the prediction of $\mathbf{x}_{\text{query}}$. Additionally, because the prediction is highly likely to be correct in the zero-shot regime (i.e., $N = 0$), the prediction in the test phase can be viewed as a competition between model memorization and adversarial in-context learning. This dynamic is primarily influenced by the number of hijacked context examples.

### 3.2. Transformer model

Following the extensive prior theoretical works for transformer (Zhang et al., 2024a;b; Chen et al., 2024; Frei & Vardi, 2024; Ahn et al., 2023), we consider linear attention-only transformers, a prevalent simplified structure to investigate the behavior of transformer models. In particular, We define an $L$-layer linear transformer TF as a stack of $L$ single-head linear attention-only layers. For the input matrix $\mathbf{Z}_{i-1} \in \mathbb{R}^{(d+1)\times(n+1)}$, the $i$-th single-head linear attention-only layer $\text{TF}_i$ updates the input as follows:

$$\mathbf{Z}_i = \text{TF}_i(\mathbf{Z}_{i-1}) = \mathbf{Z}_{i-1} + \mathbf{P}_i \mathbf{Z}_i \mathbf{M}(\mathbf{Z}_{i-1}^\top \mathbf{Q}_i \mathbf{Z}_{i-1}), \tag{3.3}$$

where $\mathbf{M} := \begin{pmatrix} \mathbf{I}_n & 0 \\ 0 & 0 \end{pmatrix} \in \mathbb{R}^{(n+1)\times(n+1)}$ is the mask matrix. We design this architecture to constrain the model's focus to the first $n$ in-context examples. Moreover, the matrix $\mathbf{P} := \mathbf{W}_v \in \mathbb{R}^{(d+1)\times(d+1)}$ serves as the value matrix in the standard self-attention layer, while the matrix $\mathbf{Q} := \mathbf{W}_k^\top \mathbf{W}_q \in \mathbb{R}^{(d+1)\times(d+1)}$ consolidates the key matrix and query matrix. This mild re-parameterization has been widely considered in numerous recent theoretical works (Huang et al., 2023; Wang et al., 2024; Tian et al., 2023; Jelassi et al., 2022). To adapt the transformer for solving the linear classification problem, we introduce an additional linear embedding layer $\mathbf{W}_E \in \mathbb{R}^{(d+1)\times(d+1)}$. Then the output of the transformer TF is defined as

$$\begin{aligned} \widehat{y}_{\text{query}} &= \text{TF}(\mathbf{Z}_0; \mathbf{W}_E, \{\mathbf{P}_\ell, \mathbf{Q}_\ell\}_{\ell=1}^L) \\ &= -[\text{TF}_L \circ \cdots \circ \text{TF}_1 \circ \mathbf{W}_E(\mathbf{Z}_0)]_{(d+1),(n+1)} \\ &= -[\mathbf{Z}_L]_{(d+1),(n+1)}, \end{aligned} \tag{3.4}$$

i.e. the negative of the $(d+1, n+1)$-th entry of $\mathbf{Z}_L$, and this position is replaced by 0 in the input $\mathbf{Z}_0$. The reason for taking the minus sign here is to align with previous work (Von Oswald et al., 2023; Shen et al., 2024), which will be explained in Proposition 4.1.

### 3.3. Evaluation metrics

Based on the illustration regarding the transformer architecture, we first define the in-context learning risk of a $L$-layer

model TF in the training phase. In particular, let $\mathcal{D}_{\mathrm{tr}}$ be the distribution of the input data matrix $\mathbf{Z}$ in (3.1) and the target $y_{\mathrm{query}}$, which covers the randomness of both $(\mathbf{x}, y)$ and $\mathbf{w}^*$, then the risk function in the training phase is defined as:

$$\mathcal{R}(\mathtt{TF}) := \mathbb{E}_{\mathbf{Z}, y_{\mathrm{query}} \sim \mathcal{D}_{\mathrm{tr}}} \left[ (\mathtt{TF}(\mathbf{Z}; \boldsymbol{\theta}) - y_{\mathrm{query}})^2 \right]. \quad (3.5)$$

where $\boldsymbol{\theta} = \{\mathbf{W}_E, \{\mathbf{P}_\ell, \mathbf{Q}_\ell\}_{\ell=1}^L\}$ denotes the collection of all trainable parameters of TF. This risk function will be leveraged for training the transformer models (where we use the stochastic gradient in the experiments).

Additionally, in the test phase, let $\mathcal{D}_{\mathrm{te}}$ be the distribution of the input data matrix $\mathbf{Z}^{\mathrm{hc}}$ in (3.2) and the target $y_{\mathrm{query}}$, we consider the following population prediction error:

$$\mathcal{E}(\mathtt{TF}) := \mathbb{P}_{\mathbf{Z}^{\mathrm{hc}}, y_{\mathrm{query}} \sim \mathcal{D}_{\mathrm{te}}} \left[ \mathtt{TF}(\mathbf{Z}^{\mathrm{hc}}; \boldsymbol{\theta}) \cdot y_{\mathrm{query}} < 0 \right]. \quad (3.6)$$

## 4. Main theory

In this section, we present how we establish our theoretical analysis framework regarding the robustness of transformers against context hijacking. In summary, we can briefly sketch our framework into the following several steps:

- Step 1. We establish the equivalence between the $L$-layer transformers and $L$ steps gradient descent, converting the original problem of identifying well-trained transformers to the problem of finding the optimal parameters of gradient descent (i.e., initialization and learning rates).
- Step 2. We derive the optimal learning rates and initialization of gradient descent, revealing its relationship with the number of layers $L$ and training context length $n$.
- Step 3. By formulating the classification error of a linear model obtained by $L$ steps gradient descent with optimal parameters on hijacking distribution $\mathcal{D}_{\mathrm{te}}$, we characterize how the number of layers $L$, the training context length $n$ and test context length $N$ affect the robustness.

### 4.1. Optimizing over in-context examples

Inspired by a line of recent works (Zhang et al., 2024b; Bai et al., 2024; Chen et al., 2024; Ahn et al., 2023; Olsson et al., 2022) which connects the in-context learning of transformer with the gradient descent algorithm, we follow a similar approach by showing that, in the following proposition, multi-layer transformer can implement multi-step gradient descent, starting from any initialization, on context examples.

**Proposition 4.1.** *For any L-layer single-head linear transformer, let $\widehat{y}_{\mathrm{query}}^{(l)}$ be the output of the l-th layer of t, i.e. the $(d+1, n+1)$-th entry of $\mathbf{Z}_l$. Then, there exists a single-head linear transformer with L layers such that $\widehat{y}_{\mathrm{query}}^{(l)} = -\langle \mathbf{w}_{\mathrm{gd}}^{(l)}, \mathbf{x}_{\mathrm{query}} \rangle$. Here, $\mathbf{w}_{\mathrm{gd}}^{(l)}$'s are the parameter*

*vectors obtained by the following gradient descent iterative rule and the initialization $\mathbf{w}_{\mathrm{gd}}^{(0)}$ can be arbitrary:*

$$\mathbf{w}_{\mathrm{gd}}^{(l+1)} = \mathbf{w}_{\mathrm{gd}}^{(l)} - \boldsymbol{\Gamma}_l \nabla \widetilde{L}(\mathbf{w}_{\mathrm{gd}}^{(l)}),$$

$$where \quad \widetilde{L}(\mathbf{w}) = \frac{1}{2} \sum_{i=1}^n (\langle \mathbf{w}, \mathbf{x}_i \rangle - y_i)^2. \quad (4.1)$$

*Here $\boldsymbol{\Gamma}_l$ can be any $d \times d$ matrix.*

As $\boldsymbol{\Gamma}_l$ could be any $d \times d$ matrix, Proposition 4.1 demonstrates that the output of the $L$-layer transformers is equivalent to that of a linear model trained via $L$-steps of full-batch preconditioned gradient descent on the context examples, with $\{\boldsymbol{\Gamma}_l\}_{l=0}^{L-1}$ being the learning rates. This suggests that each $L$-layer transformer defined in (3.4), with different parameters, can be viewed as an optimization process of a linear model characterized by a distinct set of initialization and learning rates $\{\mathbf{w}_{\mathrm{gd}}^{(0)}, \boldsymbol{\Gamma}_0, \ldots, \boldsymbol{\Gamma}_{L-1}\}$. Therefore, it suffice to directly find the optimal parameters of the gradient descent process, without needing to infer the specific parameters of the well-trained transformers.

Among all related works presenting similar conclusions that transformers can implement gradient descent, our result is general as we prove that transformers can implement multi-step gradient descent from any initialization. In comparison, for example, Zhang et al. (2024b) shows that a single-layer transformer with MLP can implement one-step gradient descent from non-zero initialization. Ahn et al. (2023) demonstrate that linear transformers can implement gradient descent, but only from $\mathbf{0}$ initialization.

### 4.2. Optimal multi-step gradient descent

Based on the discussion in the previous section, Proposition 4.1 successfully transforms the original problem of identifying the parameters of well-trained transformers into the task of finding the optimal learning rates and initialization for the gradient descent process (4.1). In this section, we present our conclusions regarding these optimal parameters. As we consider optimizing over the general training distribution $\mathcal{D}_{\mathrm{tr}}$, where the tokens $\mathbf{x}_i$'s follow the isotropic distribution, it follows that the updating step size should be equal in each direction from the perspective of expectation. Therefore we consider the case $\boldsymbol{\Gamma}_l = \alpha_l \mathbf{I}_d$ to simplify the problem, with $\alpha_l$ being a scalar for all $l \in \{0, \ldots, L-1\}$. In the following, we focus on the optimal set of parameters $\{\mathbf{w}_{\mathrm{gd}}^{(0)}, \alpha_0, \ldots, \alpha_{L-1}\}$. Specifically, we consider the population loss for $\mathbf{w}_{\mathrm{gd}}^{(L)}$ as

$$\mathcal{R}(\mathbf{w}_{\mathrm{gd}}^{(L)}) := \mathbb{E}_{T, \mathbf{w}^* \sim \mathcal{D}_{\mathrm{tr}}} \left[ \left( \langle \mathbf{w}_{\mathrm{gd}}^{(L)}, \mathbf{x}_{\mathrm{query}} \rangle - y_{\mathrm{query}} \right)^2 \right],$$

where $T = \{(\mathbf{x}_1, y_1), \ldots, (\mathbf{x}_n, y_n), (\mathbf{x}_{\text{query}}, y_{\text{query}})\}$ is the set of all classification pairs [1]. This definition resembles $\mathcal{R}(\text{TF})$ defined in (3.5). We attempt to find the $\mathbf{w}_{\text{gd}}^{(L)}$ that minimizes this population loss, along with the corresponding learning rates $\{\alpha_l\}_{l=0}^{L}$ and initialization $\mathbf{w}_{\text{gd}}^{(0)}$, which can generate this $\mathbf{w}$ via gradient descent. We first present the following proposition demonstrating that there exists commutative invariance among the learning rates $\{\alpha_l\}_{l=0}^{L}$ for producing $\mathbf{w}_{\text{gd}}^{(L)}$.

**Proposition 4.2.** *Let $\{\alpha_0, \alpha_1, \ldots, \alpha_{L-1}\}$ be a set of learning rates, and $\{\alpha_0', \alpha_1', \ldots, \alpha_{L-1}'\}$ be another set of learning rates that is a permutation of $\{\alpha_0, \alpha_1, \ldots, \alpha_{L-1}\}$, meaning both sets contain the same elements, with the only difference being the order of these elements. With $\mathbf{w}_{\text{gd}}^{(L)} \in \mathbb{R}^d$ denoting as the parameters achieved by learning rates $\{\alpha_0, \alpha_1, \ldots, \alpha_{L-1}\}$ and $\mathbf{w}_{\text{gd}'}^{(L)} \in \mathbb{R}^d$ as the parameters achieved by learning rates $\{\alpha_0', \alpha_1', \ldots, \alpha_{L-1}'\}$ from the same initialization $\mathbf{w}_{\text{gd}}^{(0)}$, it holds that $\mathbf{w}_{\text{gd}}^{(L)} = \mathbf{w}_{\text{gd}'}^{(L)}$.*

Proposition 4.2 implies that the learning rates at different steps contribute equally to the overall optimization process. Consequently, we will consider a consistent learning rate $\alpha$ through the entire gradient descent procedure, which significantly reduces the difficulty of analysis and does not incur any loss of generality. Now we are ready to present our main results regarding the derivation of the optimal parameters $\alpha$ and $\mathbf{w}_{\text{gd}}^{(0)}$.

**Theorem 4.3.** *For training distribution $\mathcal{D}_{\text{tr}}$ in Definition 3.1, suppose that the training context length $n$ is sufficiently large such that $n \geq \widetilde{\Omega}(\max\{d^2, dL\})$. Additionally, suppose that the perturbation of $\mathbf{w}^*$ around its expectation $\boldsymbol{\beta}^*$ is smaller than $\frac{\pi}{2}$, i.e. $\langle \mathbf{w}^*, \boldsymbol{\beta}^* \rangle > 0$. Based on these assumptions, the optimal learning rate $\alpha$ and initialization $\mathbf{w}_{\text{gd}}^{(0)}$, i.e. $\alpha, \mathbf{w}_{\text{gd}}^{(0)} = \arg\min_{\alpha, \mathbf{w}_{\text{gd}}^{(0)}} \mathcal{R}(\mathbf{w}_{\text{gd}}^{(L)})$, take the value as follows:*

$$\alpha = \widetilde{\Theta}\left(\frac{1}{nL}\right); \qquad \mathbf{w}_{\text{gd}}^{(0)} = c\boldsymbol{\beta}^*,$$

*where $c$ is an absolute constant.*

Theorem 4.3 clearly identifies the optimal learning rate $\alpha$ and initialization $\mathbf{w}_{\text{gd}}^{(0)}$. Specifically, it shows that the optimal initialization $\mathbf{w}_{\text{gd}}^{(0)}$ aligns the direction of the expectation $\boldsymbol{\beta}^*$, with its length independent of the number of steps $L$, and the context length $n$. Such a conclusion complies with our intuitions as the initialization $\mathbf{w}_{\text{gd}}^{(0)}$ represents the memory of large language models, which is not dependent on the task-specific context examples. In contrast, the optimal learning rate $\alpha$ is inversely related to both $n$ and $L$. This suggests that in both cases: (i) with more in-context examples;

and (ii) with more layers, the output of pre-trained transformers will equal to that of a more fine-grained gradient descent process using a smaller learning rate. Generally, a small-step strategy ensures the convergence of the objective, highlighting the potential benefits of deeper architectures and training inputs with longer context.

### 4.3. Robustness against context hijacking

The previous two subsections illustrate that for any input with context examples, we can obtain the corresponding prediction for that input from the well-trained transformers by applying gradient descent with the optimal parameters we derived in Theorem 4.3. As we model $\mathcal{D}_{\text{te}}$ the distribution of hijacking examples, to examine the robustness of $L$-layer transformers against hijacking, we only need to check whether the linear model achieved by $L$-step gradient on $(\mathbf{x}_{\text{hc}}, y_{\text{hc}})$ can still conduct successful classification on $\mathbf{x}_{\text{query}}$. Specifically, we consider the classification error of the parameter vector $\widetilde{\mathbf{w}}_{\text{gd}}^{(L)}$ as,

$$\mathcal{E}(\widetilde{\mathbf{w}}_{\text{gd}}^{(L)}) := \mathbb{P}_{T, \mathbf{w}^* \sim \mathcal{D}_{\text{te}}}\left(y_{\text{query}} \cdot \langle \widetilde{\mathbf{w}}_{\text{gd}}^{(L)}, \mathbf{x}_{\text{query}} \rangle < 0\right),$$

where $T = \{(\mathbf{x}_{\text{hc}}, y_{\text{hc}}), (\mathbf{x}_{\text{query}}, y_{\text{query}})\}$, and $\widetilde{\mathbf{w}}_{\text{gd}}^{(L)}$ is obtained by implementing gradient descent on $(\mathbf{x}_{\text{hc}}, y_{\text{hc}})$ with $L$ steps and the optimal $\alpha$ and $\mathbf{w}_{\text{gd}}^{(0)}$. Similar to the previous result, $\mathcal{E}(\widetilde{\mathbf{w}}_{\text{gd}}^{(L)})$ is identical to $\mathcal{E}(\text{TF})$ defined in (3.6). Based on these preliminaries, we are ready to present our results regarding the robustness against context hijacking. We first introduce the following lemma illustrating that when the context length of hijacking examples is small, we hardly observe the label flipping phenomenons of the prediction from well-trained transformers.

**Lemma 4.4.** *Assume that all assumptions in Theorem 4.3 still hold. Additionally, assume that the length of hijacking examples $N$ is small such that $N \leq \widetilde{O}\left(\frac{n}{d^{3/2}}\right)$ and $\sigma$ follows any continuous distribution. Based on these assumptions, it holds that*

$$\mathcal{E}(\widetilde{\mathbf{w}}_{\text{gd}}^{(L)}) \leq \mathcal{E}(\mathbf{w}_{\text{gd}}^{(0)}) + o(1).$$

Lemma 4.4 demonstrates that when the context length of hijacking examples is small, the classification error of the linear model obtained through gradient descent on these hijacking examples is very close to that of the optimal initialization. The reasoning is straightforward: When $N$ is relatively smaller compared to training context length $n$, and since the optimal learning rate $\alpha$ is on the order of the reciprocal of $n$, the contributions from the hijacking examples become almost negligible in gradient descent iterations, allowing the model to remain close to its initialization. Consequently, we consider the case that $N$ is comparable with $n$ in the following theorem.

---

[1] Here we slightly abuse the notation of $\mathcal{D}_{\text{tr}}$ to denote both the distribution of $\mathbf{Z}$, $\mathbf{w}_{\text{query}}$ and $T$, $\mathbf{w}^*$.

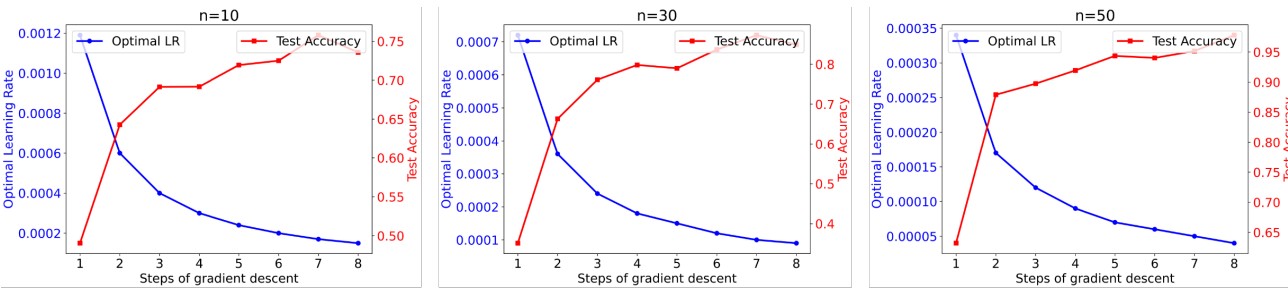

*Figure 2.* Gradient descent experiments using a single-layer neural network. We use grid search to obtain the optimal learning rate for different training context lengths $n$ and different steps of gradient descent $L$. Then we use the corresponding optimal learning rate to perform multi-step gradient descent optimization on the test dataset. The results show that longer training context lengths and more gradient descent steps lead to smaller optimal learning rate and better optimization.

**Theorem 4.5.** *Assume that all assumptions in Theorem 4.3 still hold. Additionally, assume that $N \geq \widetilde{\Omega}\left(\frac{n}{d^{3/2}}\right)$, $n \geq \widetilde{\Omega}(Nd)$, and $\sigma$ follows some uniform distribution. Based on these assumptions, it holds that*

$$\mathcal{E}(\widetilde{\mathbf{w}}_{\mathrm{gd}}^{(L)}) \leq c_1 - c_2 \left(1 - \widetilde{\Theta}\left(\frac{Nd}{nL}\right)\right)^L,$$

*where $c_1$, $c_2$ are two positive scalar solely depending on the distribution of $\sigma$ and $\mathbf{w}^*$.*

Based on a general assumption that $\sigma$ follows the uniform distribution, Theorem 4.5 formulates the upper bound of the classification error as a function of the training context length $n$, the number of hijacking examples $N$, and the number of layers $L$. Specifically, this upper bound contains a term proportional to $-\left(1 - \widetilde{\Theta}\left(\frac{Nd}{nL}\right)\right)^L$. As $\left(1 - \widetilde{\Theta}\left(\frac{Nd}{nL}\right)\right)^L$ is a monotonically increasing function for $N$ and a monotonically decreasing function for $n$ and $L$, Theorem 4.5 successfully demonstrates two facts: (i) well-trained transformers with deeper architectures, or those pre-trained on longer context examples, will exhibit more robustness against context hijacking; and (ii) for a given well-trained transformer, the context hijacking phenomenon is easier to observe when provided with more hijacking examples. These conclusions align well with our experimental observations (Figure 3).

## 5. Experiments

In this section, we conduct experiments based on the setting in Section 3 to verify the theory we proposed in Section 4. We first verify the consistency of our theory with optimal multi-step gradient descent. Then we train a series of linear transformers to examine their robustness on test data.

### 5.1. Optimal gradient descent with different steps

In our theoretical framework, for the same optimization objective, the optimal gradient descent with more steps $L$ or longer training context length $n$ will have a smaller learning rate per step (Theorem 4.3), and this combination of more steps with small learning rates will perform better on the optimization process over context samples (Theorem 4.5). Our theory shows that a trained transformer will learn the optimal multi-step gradient descent, which will make it more robust during testing. Therefore, we directly verified the consistency between practice and theory in the multi-step gradient descent experiment.

We construct a single-layer neural network to conduct optimal multi-step gradient descent experiments. Each training sample $(\mathbf{x}_i, y_i)$ is drawn i.i.d. from the distribution $\mathcal{D}_{\mathrm{tr}}$ defined in Section 3.1. We consider the learning rate that minimizes the loss of the test sample which is also drawn from $\mathcal{D}_{\mathrm{tr}}$ when the single-layer neural network is trained using 1 to 8 steps of gradient descent, that is, the optimal learning rate $\alpha_L$ corresponding to $L$-step gradient descent, which can be obtained by grid search. Figure 2 shows that $\alpha_L$ decreases as $L$ and $n$ increases, which is aligned with our theoretical results (Theorem 4.3).

Next, we discuss the second part of the theoretical framework, i.e., gradient descent with more steps and small step size performs a more fine-grained optimization (Theorem 4.5), which can be verified by our experiment results. We apply the optimal learning rate searched in the training phase to the test phase, and perform gradient descent optimization on the test samples drawn from $\mathcal{D}_{\mathrm{te}}$ with the optimal learning rate and its corresponding number of steps. We can find that with the increase in the number of gradient descent steps and the decrease in the corresponding learning rate, the performance of the model will be significantly improved.

### 5.2. Robustness of linear transformers with different number of layers

Applying our theoretical framework to the context hijacking task on transformers can explain it well, indicating that our theory has practical significance. We train linear transformers with different depths and context lengths on the training dataset based on distribution $\mathcal{D}_{\mathrm{tr}}$. We mainly investigate

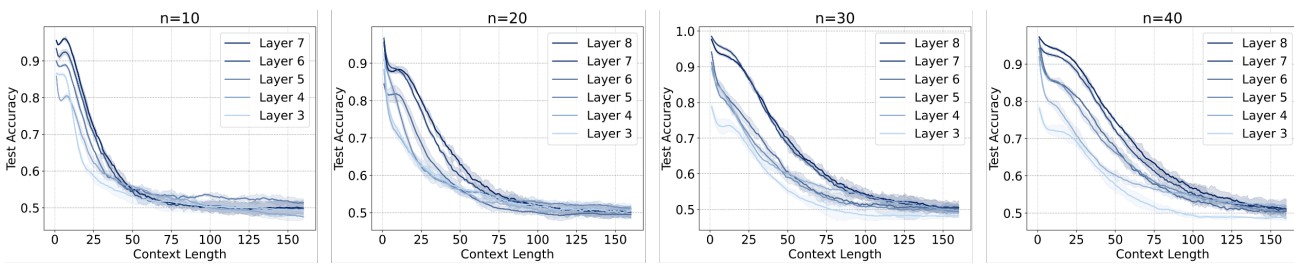

*Figure 3.* Linear transformers experiments with different depths and different training context lengths. By testing the trained linear transformers on the test set, we can find that as the number of interference samples increases, the model prediction accuracy becomes worse. However, deeper models have higher accuracy, indicating stronger robustness. As the training context length increases, the model robustness will also increase because the accuracy converges significantly more slowly.

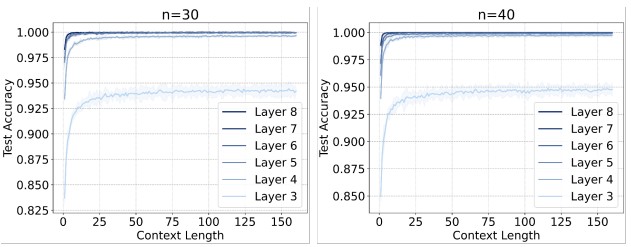

*Figure 4.* Linear transformers experiments on training dataset. By testing trained linear transformers on the training set, the initial accuracy of the model is high and can be improved with the increase of context length, indicating that the model can use in-context learning to fine-tune $\boldsymbol{\beta}^\star$ to $\mathbf{w}^\star$. And deeper models have stronger optimization capabilities.

the impact of training context length $n$, and model depth $L$ and the testing context length $N$ on model classification accuracy.

We first test the trained transformers on the training dataset to verify that the model can fine-tune the memorized $\boldsymbol{\beta}^*$ to $\mathbf{w}^\star$. According to the Figure 4, we can find that the model has a high classification accuracy when there are very few samples at the beginning. This means that the model successfully memorizes the shared signal $\boldsymbol{\beta}^*$. As the context length increases, the accuracy of the model gradually increases and converges, meaning that the model can fine-tune the pre-trained $\boldsymbol{\beta}^*$ by using the context samples. In addition, deeper models can converge to larger values faster, corresponding to the theoretical view that deeper models can perform more sophisticated optimization.

Then we conduct experiments on the test set. Observing the experiment results (Figure 3), we can see that as the context length increases, the accuracy of the model decreases significantly and converges to 50%, showing that the model is randomly classifying the final query $\mathbf{x}_{\text{query}}$. This is consistent with the context hijacking phenomenon that the model's robustness will deteriorate as the number of interference prompt words increases. When the number of layers increases, the models with different depths show the same

trend as the context length increases, but the accuracy of the model will increase significantly, which is consistent with the phenomenon that deeper models show stronger robustness in practical applications. In addition, the model becomes significantly more robust as the training context length increases, which is reflected in the fact that the classification accuracy converges more slowly as the length increases.

## 6. Conclusion and discussion

In this paper, we explore the robustness of transformers from the perspective of in-context learning. We are inspired by the real-world problem of LLMs, namely context hijacking (Jiang et al., 2024), and we build a comprehensive theoretical framework by modeling context hijacking phenomenon as a linear classification problem. We first demonstrate the context hijacking phenomenon by conducting experiments on LLMs with different depths, i.e., the output of the LLMs can be simply manipulated by modifying the context with factually correct information. This reflects an intuition: deeper models may be more robust. Then we develop a comprehensive theoretical analysis of the robustness of transformer, showing that the well-trained transformers can achieve the optimal gradient descent strategy. More specifically, we show that as the number of model layers or the length of training context increase, the model will be able to perform more fine-grained optimization steps over context samples, which can be less affected by the hijacking examples, leading to stronger robustness. Specifically considering the context hijacking task, our theory can fully explain the various phenomena, which is supported by a series of numerical experiments.

Our work provides a new perspective for the robustness explanation of transformers and the understanding of in-context learning ability, which offer new insights to understand the benefit of deeper architecture. Besides, our analysis on the optimal multi-step gradient descent may also be leveraged to other problems that involve the numerical optimization for linear problems.

## Impact Statement

This paper presents work whose goal is to advance the field of Machine Learning. There are many potential societal consequences of our work, none which we feel must be specifically highlighted here.

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

## A. Notations

Given two sequences $\{x_n\}$ and $\{y_n\}$, we denote $x_n = O(y_n)$ if there exist some absolute constant $C_1 > 0$ and $N > 0$ such that $|x_n| \le C_1|y_n|$ for all $n \ge N$. Similarly, we denote $x_n = \Omega(y_n)$ if there exist $C_2 > 0$ and $N > 0$ such that $|x_n| \ge C_2|y_n|$ for all $n > N$. We say $x_n = \Theta(y_n)$ if $x_n = O(y_n)$ and $x_n = \Omega(y_n)$ both holds. Additionally, we denote $x_n = o(y_n)$ if, for any $\epsilon > 0$, there exists some $N(\epsilon) > 0$ such that $|x_n| \le \epsilon|y_n|$ for all $n \ge N(\epsilon)$, and we denote $x_n = \omega(y_n)$ if $y_n = o(x_n)$. We use $\widetilde{O}(\cdot)$, $\widetilde{\Omega}(\cdot)$, and $\widetilde{\Theta}(\cdot)$ to hide logarithmic factors in these notations respectively. Finally, for any $n \in \mathbb{N}_+$, we use $[n]$ to denote the set $\{1, 2, \cdots, n\}$.

## B. Proof of Proposition 4.1

In this section we provide a proof for Proposition 4.1.

*Proof of Proposition 4.1.* Our proof is inspired by Lemma 1 in Ahn et al. (2023), while we consider a non-zero initialization. We first provide the parameters $\mathbf{W}_E, \mathbf{P}_\ell, \mathbf{Q}_\ell \in \mathbb{R}^{(d+1)\times(d+1)}$ of a $L$-layers transformer.

$$\mathbf{W}_E = \begin{bmatrix} \mathbf{I}_n & 0 \\ -\mathbf{w}_{\text{gd}}^{(0)} & 1 \end{bmatrix}, \quad \mathbf{P}_\ell = \begin{bmatrix} \mathbf{0}_{d\times d} & 0 \\ 0 & 1 \end{bmatrix}, \quad \mathbf{Q}_\ell = \begin{bmatrix} -\mathbf{\Gamma}_\ell & 0 \\ 0 & 0 \end{bmatrix} \quad \text{where } \mathbf{\Gamma}_\ell \in \mathbb{R}^{d\times d}.$$

For the linear classification problem, the input sample $\mathbf{Z}_0 \in \mathbb{R}^{(d+1)\times(n+1)}$ consists of $\{(\mathbf{x}_i, y_i)\}_i = 1^n$ and $(\mathbf{x}_{\text{query}}, y_{\text{query}})$ in (3.1) , which will first be embedded by $\mathbf{W}_E$. Let $\mathbf{X}^{(0)} \in \mathbb{R}^{d\times(n+1)}$ denote the first $d$ rows of $\mathbf{W}_E(\mathbf{Z}_0)$ and let $\mathbf{Y}^{(0)} \in \mathbb{R}^{1\times(n+1)}$ denote the $(d+1)$-th row of $\mathbf{W}_E(\mathbf{Z}_0)$. In subsequent iterative updates in (3.3), the values at the same position will be denoted as $\mathbf{X}^{(l)}$ and $\mathbf{Y}^{(l)}$, for $l = 1, \ldots, L$. Similarly, define $\bar{\mathbf{X}}^{(l)} \in \mathbb{R}^{d\times n}$ and $\bar{\mathbf{Y}}^{(l)} \in \mathbb{R}^{1\times n}$ as matrices that exclude the last query sample $(\mathbf{x}_{\text{query}}^{(l)}, y_{\text{query}}^{(l)})$. That is, they only contain the first $n$ columns of the output of the $l$-th layer. Let $\mathbf{x}_i^{(l)}$ and $y_i^{(l)}$ be the $i$-th pair of samples output by the $l$-th layer. Define a function $g(\mathbf{x}, y, l) : \mathbb{R}^d \times \mathbb{R} \times \mathbb{Z} \to \mathbb{R}$: let $\mathbf{x}_{\text{query}}^{(0)} = \mathbf{x}$ and $y_{\text{query}}^{(0)} = y - \langle \mathbf{w}_{\text{gd}}^{(0)}, \mathbf{x}\rangle$, then $g(\mathbf{x}, y, l) := y_{\text{query}}^{(k)}$. Next, based on the update formula (3.3) and the parameters constructed above, we have:

$$\mathbf{X}^{(l+1)} = \mathbf{X}^{(l)} = \cdots = \mathbf{X}^{(0)}, \quad \mathbf{Y}^{(l+1)} = \mathbf{Y}^{(l)} - \mathbf{Y}^{(l)}\mathbf{M}(\mathbf{X}^{(0)})^\top \mathbf{\Gamma}_l \mathbf{X}^{(0)}.$$

Then for all $i \in \{1, \ldots, n\}$,

$$y_i^{(l+1)} = y_i^{(l)} - \sum_{j=1}^n \mathbf{x}_i^\top \mathbf{\Gamma}_l \mathbf{x}_j y_j^{(l)}.$$

So $y_i^{(l+1)}$ does not depend on $y_{\text{query}}^{(l+1)}$. For query position,

$$y_{\text{query}}^{(l+1)} = y_{\text{query}}^{(l)} - \sum_{j=1}^n \mathbf{x}_{\text{query}}^\top \mathbf{\Gamma}_l \mathbf{x}_j y_j^{(l)}.$$

Then we obtain $g(\mathbf{x}, y, l)$ and $g(\mathbf{x}, 0, l)$:

$$\begin{aligned}
g(\mathbf{x}, y, l) &= y_{\text{query}}^{(l-1)} - \sum_{j=1}^n \mathbf{x}_{\text{query}}^\top \mathbf{\Gamma}_{l-1} \mathbf{x}_j y_j^{(l-1)} \\
&= y_{\text{query}}^{(l-2)} - \sum_{j=1}^n \mathbf{x}_{\text{query}}^\top \mathbf{\Gamma}_{l-2} \mathbf{x}_j y_j^{(l-2)} - \sum_{j=1}^n \mathbf{x}_{\text{query}}^\top \mathbf{\Gamma}_{l-1} \mathbf{x}_j y_j^{(l-1)} \\
&\quad\vdots \\
&= y_{\text{query}}^{(0)} - \sum_{j=1}^n \mathbf{x}_{\text{query}}^\top \mathbf{\Gamma}_0 \mathbf{x}_j y_j^{(0)} - \cdots - \sum_{j=1}^n \mathbf{x}_{\text{query}}^\top \mathbf{\Gamma}_{l-1} \mathbf{x}_j y_j^{(l-1)} \\
&= y - \langle \mathbf{w}_{\text{gd}}^{(0)}, \mathbf{x}_{\text{query}}\rangle - \sum_{j=1}^n \mathbf{x}_{\text{query}}^\top \mathbf{\Gamma}_0 \mathbf{x}_j y_j^{(0)} - \cdots - \sum_{j=1}^n \mathbf{x}_{\text{query}}^\top \mathbf{\Gamma}_{l-1} \mathbf{x}_j y_j^{(l-1)};
\end{aligned}$$

$$g(\mathbf{x}, 0, k) = y_{\text{query}}^{(l-1)} - \sum_{j=1}^{n} \mathbf{x}_{\text{query}}^{\top} \mathbf{\Gamma}_{l-1} \mathbf{x}_j y_j^{(l-1)}$$

$$= y_{\text{query}}^{(l-2)} - \sum_{j=1}^{n} \mathbf{x}_{\text{query}}^{\top} \mathbf{\Gamma}_{l-2} \mathbf{x}_j y_j^{(l-2)} - \sum_{j=1}^{n} \mathbf{x}_{\text{query}}^{\top} \mathbf{\Gamma}_{l-1} \mathbf{x}_j y_j^{(l-1)}$$

$$\vdots$$

$$= y_{\text{query}}^{(0)} - \sum_{j=1}^{n} \mathbf{x}_{\text{query}}^{\top} \mathbf{\Gamma}_0 \mathbf{x}_j y_j^{(0)} - \cdots - \sum_{j=1}^{n} \mathbf{x}_{\text{query}}^{\top} \mathbf{\Gamma}_{l-1} \mathbf{x}_j y_j^{(l-1)}$$

$$= -\langle \mathbf{w}_{\text{gd}}^{(0)}, \mathbf{x}_{\text{query}} \rangle - \sum_{j=1}^{n} \mathbf{x}_{\text{query}}^{\top} \mathbf{\Gamma}_0 \mathbf{x}_j y_j^{(0)} - \cdots - \sum_{j=1}^{n} \mathbf{x}_{\text{query}}^{\top} \mathbf{\Gamma}_{l-1} \mathbf{x}_j y_j^{(l-1)}; .$$

So we have $g(\mathbf{x}, y, l) = g(\mathbf{x}, 0, l) + y$. Observing $g(\mathbf{x}, 0, l)$, we can find that it is linear in $\mathbf{x}$ for the reason that every term of $g(\mathbf{x}, 0, l)$ is linear in $\mathbf{x}_{\text{query}}$, which means we can rewrite it. We verify that there exists a $\boldsymbol{\theta}_l \in \mathbb{R}^d$ for each $l \in [L]$, such that for all $\mathbf{x}, y$,

$$g(\mathbf{x}, y, l) = g(\mathbf{x}, 0, k) + y = \langle \boldsymbol{\theta}_l, \mathbf{x} \rangle + y.$$

Let $l = 0$, we have $\langle \boldsymbol{\theta}_0, \mathbf{x} \rangle = g(\mathbf{x}, y, 0) - y = y_{\text{query}}^{(0)} - y = -\langle \mathbf{w}_{\text{gd}}^{(0)}, \mathbf{x}_{\text{query}} \rangle$, so $\boldsymbol{\theta}_0 = -\mathbf{w}_{\text{gd}}^{(0)}$. Next, we will show that for all $(\mathbf{x}_i, y_i) \in \{(\mathbf{x}_1, y_1) \ldots, (\mathbf{x}_n, y_n), (\mathbf{x}_{\text{query}}, y_{\text{query}})\}$,

$$g(\mathbf{x}_i, y_i, l) = y_i^{(l)} = \langle \boldsymbol{\theta}_l, \mathbf{x}_i \rangle + y_i.$$

Observing the update formulas for $y_i^{(l+1)}$ and $y_{\text{query}}^{(l+1)}$, if we let $\mathbf{x}_{\text{query}} := \mathbf{x}_i$ for some $i$, we can get that $y_i^{(l+1)} = y_{\text{query}}^{(l+1)}$ because $y_i^{(0)} = y_{\text{query}}^{(0)}$ by definition. This indicates that

$$\bar{\mathbf{Y}}^{(l)} = \bar{\mathbf{Y}}^{(0)} + \boldsymbol{\theta}_l^T \bar{\mathbf{X}}.$$

Finally, we can rewrite the update formula for $y_k^{(n+1)}$

$$y_{\text{query}}^{(l+1)} = y_{\text{query}}^{(l)} - \sum_{j=1}^{n} \mathbf{x}_{\text{query}}^{\top} \mathbf{\Gamma}_l \mathbf{x}_j y_j^{(l)}.$$

$$= y_{\text{query}}^{(l)} - \langle \mathbf{\Gamma}_l \bar{\mathbf{X}} (\bar{\mathbf{Y}}^{(l)})^{\top}, \mathbf{x}_{\text{query}} \rangle$$

$$\Rightarrow \quad \langle \boldsymbol{\theta}_{l+1}, \mathbf{x}_{\text{query}} \rangle = \langle \boldsymbol{\theta}_l, \mathbf{x}_{\text{query}} \rangle - \langle \mathbf{\Gamma}_l \bar{\mathbf{X}} (\bar{\mathbf{X}}^{\top} \boldsymbol{\theta}_l + (\bar{\mathbf{Y}}^{(0)})^{\top}), \mathbf{x}_{\text{query}} \rangle$$

Since $\mathbf{x}_{\text{query}}$ is an arbitrary variable, we get the more general update formula for $\boldsymbol{\theta}_l$:

$$\boldsymbol{\theta}_{l+1} = \boldsymbol{\theta}_l - \langle \mathbf{\Gamma}_l \bar{\mathbf{X}} \left( \bar{\mathbf{X}}^{\top} \boldsymbol{\theta}_l + (\bar{\mathbf{Y}}^{(0)})^{\top} \right) \rangle.$$

Notice that we use the mean squared error, we have

$$\widetilde{L}(\mathbf{w}) = \frac{1}{2} \sum_{i=1}^{n} (\langle \mathbf{w}, \mathbf{x}_i \rangle - y_i)^2$$

$$= \frac{1}{2} \|\bar{\mathbf{X}}^{\top} \mathbf{w} - (\bar{\mathbf{Y}}^{(0)})^{\top}\|_2.$$

Then we get its gradient $\nabla \widetilde{L}(\mathbf{w}) = \bar{\mathbf{X}} \left( \bar{\mathbf{X}}^{\top} \mathbf{w} - (\bar{\mathbf{Y}}^{(0)})^{\top} \right)$. Let $\mathbf{w}_{\text{gd}}^{(l)} := -\boldsymbol{\theta}_l$, we have

$$\boldsymbol{\theta}_{l+1} = \boldsymbol{\theta}_l - \langle \mathbf{\Gamma}_l \bar{\mathbf{X}} \left( \bar{\mathbf{X}}^{\top} \boldsymbol{\theta}_l + \bar{\mathbf{Y}}_0^{\top} \right) \rangle$$

$$\Rightarrow \quad \mathbf{w}_{\text{gd}}^{(l+1)} = \mathbf{w}_{\text{gd}}^{(l)} - \langle \mathbf{\Gamma}_k \bar{\mathbf{X}} \left( \bar{\mathbf{X}}^{\top} \mathbf{w}_{\text{gd}}^{(l)} - (\bar{\mathbf{Y}}^{(0)})^{\top} \right) \rangle$$

$$= \mathbf{w}_{\text{gd}}^{(l)} - \mathbf{\Gamma}_l \nabla \widetilde{L}(\mathbf{w}_{\text{gd}}^{(l)}).$$

And the output of the $l$-th layer $y_{\text{query}}^{(l)}$ is

$$g\left(\mathbf{x}_{\text{query}}, y_{\text{query}}, l\right) = y_{\text{query}} + \langle \boldsymbol{\theta}_l, \mathbf{x}_{\text{query}} \rangle = y_{\text{query}} - \langle \mathbf{w}_{\text{gd}}^{(l)}, \mathbf{x}_{\text{query}} \rangle.$$

In our settings, we have $y_k^{n+1} = -\langle \mathbf{w}_{\text{gd}}^{(l)}, \mathbf{x}_{\text{query}} \rangle$ because the input query label is 0. □

## C. Gradient descent updates of parameters

In this section, we provide further details regarding the updating of parameters $\mathbf{w}_{\text{gd}}^{(l)}$, which will be utilized in subsequent proof. Besides, it can directly imply Proposition 4.2. Before demonstrating the mathematical, we first introduce several utility notations, which will be used in subsequent technical derivations and proofs. We denote $\mathcal{S}_{l,k}$ as the set of all $k$-dimensional tuples whose entries are drawn from $\{0, 1, \ldots, l-1\}$ without replacement, i.e.

$$\mathcal{S}_{t,k} = \left\{(j_1, j_2, \ldots, j_k) | j_1, j_2, \ldots, j_k \in \{0, 1, \ldots, l-1\}; j_1 \neq j_2 \neq \ldots \neq j_k\right\}.$$

Then given the set of all historical learning rates before or at $l$-th iteration, i.e. $\{\alpha_0, \alpha_1, \ldots, \alpha_{l-1}\}$, and $\mathcal{S}_{l,k}$ defined above, we define $A_{l,k}$ as

$$A_{l,k} := \sum_{(j_1, j_2, \ldots, j_k) \in \mathcal{S}_{l,k}} \prod_{\kappa=1}^{k} \alpha_{j_\kappa}.$$

Then we can observe that the permutation of elements of $\{\alpha_0, \alpha_1, \ldots, \alpha_{l-1}\}$ would not change the value of $A_{l,k}$. Then based on these notations, we present mathematical derivation in the following.

By some basic gradient calculations, we can re-write the iterative rule of gradient descent (4.1) as

$$\mathbf{w}_{\text{gd}}^{(l+1)} = \mathbf{w}_{\text{gd}}^{(l)} - \alpha_l \nabla L(\mathbf{w}_{\text{gd}}^{(l)})$$

$$= \mathbf{w}_{\text{gd}}^{(l)} - \alpha_l \sum_{i=1}^{n} \left(\langle \mathbf{w}_{\text{gd}}^{(l)}, \mathbf{x}_i \rangle - \text{sign}(\langle \mathbf{w}^*, \mathbf{x}_i \rangle)\right) \cdot \mathbf{x}_i$$

$$= \left(\mathbf{I}_d - \alpha_l \left(\sum_{i=1}^{n} \mathbf{x}_i \mathbf{x}_i^\top\right)\right) \cdot \mathbf{w}_{\text{gd}}^{(l)} + \alpha_l \left(\sum_{i=1}^{n} \text{sign}(\langle \mathbf{w}^*, \mathbf{x}_i \rangle) \cdot \mathbf{x}_i\right). \quad (\text{C.1})$$

Based on this detailed iterative formula, and the definition of $\mathcal{S}_{l,k}$ and $A_{l,k}$ above, we present and prove the following lemma, which characterizes the closed-form expression for $\mathbf{w}^{(l)}$.

**Lemma C.1.** *For the iterates of gradient descent, i.e. $\mathbf{w}_{\text{gd}}^{(l)}$'s with $l \in \{0, 1, \ldots, L-1\}$, it holds that*

$$\mathbf{w}_{\text{gd}}^{(l)} = \left(\mathbf{I}_d + \sum_{k=1}^{l} A_{l,k} \left(-\sum_{i=1}^{n} \mathbf{x}_i \mathbf{x}_i^\top\right)^k\right) \cdot \mathbf{w}_{\text{gd}}^{(0)} + \left(\sum_{k=1}^{l} A_{l,k} \left(-\sum_{i=1}^{n} \mathbf{x}_i \mathbf{x}_i^\top\right)^{k-1}\right) \cdot \left(\sum_{i=1}^{n} \text{sign}(\langle \mathbf{w}^*, \mathbf{x}_i \rangle) \cdot \mathbf{x}_i\right). \quad (\text{C.2})$$

*Proof of Lemma C.1.* Before we demonstrate our proof, we first present some conclusions regarding $\mathcal{S}_{l,k}$ and $A_{l,k}$. By directly applying the Binomial theorem and the definition of $A_{l,k}$, we can obtain that

$$\prod_{k=0}^{l-1} \left(\mathbf{I}_d + \alpha_k \left(-\sum_{i=1}^{n} \mathbf{x}_i \mathbf{x}_i^\top\right)\right) = \mathbf{I}_d + \sum_{k=1}^{l} A_{l,k} \left(-\sum_{i=1}^{n} \mathbf{x}_i \mathbf{x}_i^\top\right)^k. \quad (\text{C.3})$$

Additionally, by utilizing the definition of $\mathcal{S}_{l,k}$, we can easily derive that

$$\begin{aligned}
\mathcal{S}_{l+1,k} &= \left\{(j_1, j_2, \ldots, j_k) | j_1, j_2, \ldots, j_k \in \{0, 1, \ldots, l\}; j_1 \neq j_2 \neq \ldots \neq j_k\right\} \\
&= \left\{(j_1, j_2, \ldots, j_k) | j_1, j_2, \ldots, j_k \in \{0, 1, \ldots, l-1\}; j_1 \neq j_2 \neq \ldots \neq j_k\right\} \\
&\quad \cup \left\{(j_1, j_2, \ldots, j_{k-1}, l) | j_1, j_2, \ldots, j_{k-1} \in \{0, 1, \ldots, l-1\}; j_1 \neq j_2 \neq \ldots \neq j_{k-1}\right\} \\
&= \mathcal{S}_{l,k} \cup \left\{(j_1, j_2, \ldots, j_{k-1}, l) | (j_1, j_2, \ldots, j_{k-1}) \in \mathcal{S}_{l,k-1}\right\},
\end{aligned}$$

holds when $k \leq l$. This result can further imply that

$$A_{l+1,k} = \sum_{(j_1,j_2,\ldots,j_k) \in \mathcal{S}_{l+1,k}} \prod_{\kappa=1}^{k} \alpha_{j_\kappa} = \sum_{(j_1,j_2,\ldots,j_k) \in \mathcal{S}_{l,k}} \prod_{\kappa=1}^{k} \alpha_{j_\kappa} + \sum_{(j_1,j_2,\ldots,j_{k-1}) \in \mathcal{S}_{l,k-1}} \prod_{\kappa=1}^{k-1} \alpha_{j_\kappa} \alpha_l = A_{l,k} + \alpha_l A_{l,k-1}.$$

(C.4)

holds when $k \leq l$. Additionally, it is straightforward that

$$A_{l+1,l+1} = \alpha_l A_{l,l}; \quad A_{l+1,1} = A_{l,1} + \alpha_l. \tag{C.5}$$

With these conclusions in hands, we will begin proving this lemma by induction. When $l = 1$, by the iterative rule (C.1), we can obtain that

$$\mathbf{w}_{\text{gd}}^{(1)} = \left(\mathbf{I}_d - \alpha_0\left(\sum_{i=1}^{n} \mathbf{x}_i \mathbf{x}_i^{\top}\right)\right) \cdot \mathbf{w}_{\text{gd}}^{(0)} + \alpha_0\left(\sum_{i=1}^{n} \text{sign}(\langle \mathbf{w}^*, \mathbf{x}_i \rangle) \cdot \mathbf{x}_i\right),$$

which follows the conclusion of (C.2) due to (C.3) and the definition of $A_{1,1}$. By induction, we assume that (C.2) holds at $l$-th iteration. Then at $(l+1)$-th iteration, we can obtain that,

$$\mathbf{w}_{\text{gd}}^{(l+1)} = \left(\mathbf{I}_d - \alpha_l\left(\sum_{i=1}^{n} \mathbf{x}_i \mathbf{x}_i^{\top}\right)\right) \cdot \mathbf{w}_{\text{gd}}^{(l)} + \alpha_l\left(\sum_{i=1}^{n} \text{sign}(\langle \mathbf{w}^*, \mathbf{x}_i \rangle) \cdot \mathbf{x}_i\right)$$

$$= \alpha_l\left(\sum_{i=1}^{n} \text{sign}(\langle \mathbf{w}^*, \mathbf{x}_i \rangle) \cdot \mathbf{x}_i\right) + \left(\mathbf{I}_d - \alpha_l\left(\sum_{i=1}^{n} \mathbf{x}_i \mathbf{x}_i^{\top}\right)\right)$$

$$\cdot \left\{\prod_{\tau=0}^{l-1}\left(\mathbf{I}_d - \alpha_\tau\left(\sum_{i=1}^{n} \mathbf{x}_i \mathbf{x}_i^{\top}\right)\right) \cdot \mathbf{w}_{\text{gd}}^{(0)} + \left(\sum_{k=1}^{t} A_{l,k}\left(-\sum_{i=1}^{n} \mathbf{x}_i \mathbf{x}_i^{\top}\right)^{k-1}\right) \cdot \left(\sum_{i=1}^{n} \text{sign}(\langle \mathbf{w}^*, \mathbf{x}_i \rangle) \cdot \mathbf{x}_i\right)\right\}$$

$$= \alpha_l\left(\sum_{i=1}^{n} \text{sign}(\langle \mathbf{w}^*, \mathbf{x}_i \rangle) \cdot \mathbf{x}_i\right) + \prod_{k=0}^{l}\left(\mathbf{I}_d - \alpha_k\left(\sum_{i=1}^{n} \mathbf{x}_i \mathbf{x}_i^{\top}\right)\right) \cdot \mathbf{w}_{\text{gd}}^{(0)}$$

$$+ \left(\sum_{k=1}^{l} A_{l,k}\left(-\sum_{i=1}^{n} \mathbf{x}_i \mathbf{x}_i^{\top}\right)^{k-1} + \sum_{k=1}^{l} \alpha_l A_{l,k}\left(-\sum_{i=1}^{n} \mathbf{x}_i \mathbf{x}_i^{\top}\right)^{k}\right) \cdot \left(\sum_{i=1}^{n} \text{sign}(\langle \mathbf{w}^*, \mathbf{x}_i \rangle) \cdot \mathbf{x}_i\right)$$

$$= \prod_{k=0}^{l}\left(\mathbf{I}_d - \alpha_k\left(\sum_{i=1}^{n} \mathbf{x}_i \mathbf{x}_i^{\top}\right)\right) \cdot \mathbf{w}_{\text{gd}}^{(0)} + (A_{l,1} + \alpha_l) \cdot \left(\sum_{i=1}^{n} \text{sign}(\langle \mathbf{w}^*, \mathbf{x}_i \rangle) \cdot \mathbf{x}_i\right)$$

$$+ \left(\sum_{k=2}^{l}(A_{l,k} + \alpha_l A_{l,k-1})\left(-\sum_{i=1}^{n} \mathbf{x}_i \mathbf{x}_i^{\top}\right)^{k-1}\right) \cdot \left(\sum_{i=1}^{n} \text{sign}(\langle \mathbf{w}^*, \mathbf{x}_i \rangle) \cdot \mathbf{x}_i\right)$$

$$+ \alpha_l A_{l,l}\left(\sum_{i=1}^{n} \mathbf{x}_i \mathbf{x}_i^{\top}\right)^{l} \cdot \left(\sum_{i=1}^{n} \text{sign}(\langle \mathbf{w}^*, \mathbf{x}_i \rangle) \cdot \mathbf{x}_i\right)$$

$$= \prod_{k=0}^{l}\left(\mathbf{I}_d - \alpha_k\left(\sum_{i=1}^{n} \mathbf{x}_i \mathbf{x}_i^{\top}\right)\right) \cdot \mathbf{w}_{\text{gd}}^{(0)} + A_{l+1,1} \cdot \left(\sum_{i=1}^{n} \text{sign}(\langle \mathbf{w}^*, \mathbf{x}_i \rangle) \cdot \mathbf{x}_i\right)$$

$$+ \left(\sum_{k=2}^{t} A_{l+1,k}\left(-\sum_{i=1}^{n} \mathbf{x}_i \mathbf{x}_i^{\top}\right)^{k-1}\right) \cdot \left(\sum_{i=1}^{n} \text{sign}(\langle \mathbf{w}^*, \mathbf{x}_i \rangle) \cdot \mathbf{x}_i\right) + A_{l+1,l+1}\left(-\sum_{i=1}^{n} \mathbf{x}_i \mathbf{x}_i^{\top}\right)^{k} \cdot \left(\sum_{i=1}^{n} \text{sign}(\langle \mathbf{w}^*, \mathbf{x}_i \rangle) \cdot \mathbf{x}_i\right)$$

$$= \left(\mathbf{I}_d + \sum_{k=1}^{l+1} A_{l+1,k}\left(-\sum_{i=1}^{n} \mathbf{x}_i \mathbf{x}_i^{\top}\right)^{k}\right) \cdot \mathbf{w}_{\text{gd}}^{(0)} + \left(\sum_{k=1}^{l+1} A_{l+1,k}\left(-\sum_{i=1}^{n} \mathbf{x}_i \mathbf{x}_i^{\top}\right)^{k-1}\right) \cdot \left(\sum_{i=1}^{n} \text{sign}(\langle \mathbf{w}^*, \mathbf{x}_i \rangle) \cdot \mathbf{x}_i\right).$$

The second equality holds by substituting $\mathbf{w}_{\text{gd}}^{(l)}$ with its expansion from (C.2), assuming it is valid at the $l$-th iteration by induction. The third and fourth equalities are established by rearranging the terms. The penultimate equality is derived by applying the conclusions regarding $A_{l,k}$ from (C.4) and (C.5). The final equality is obtained by applying (C.3). This demonstrates that (C.2) still holds at $l+1$-th iteration given it holds at $l$-th iteration, which finishes the proof of induction. $\square$

Lemma C.1 demonstrate that the learning rates $\alpha_l$'s will only influence $\mathbf{w}_{\text{gd}}^{(L)}$ by determining the value of $A_{L,k}$'s. While as we have discussed above, the values of $A_{L,k}$'s depend solely on the elements in the $\{\alpha_0, \ldots, \alpha_{L-1}\}$, and remain unchanged when the order of these learning rates is rearranged. Consequently, the permutation of $\{\alpha_0, \ldots, \alpha_{L-1}\}$ will also not affect the value of $\mathbf{w}_{\text{gd}}^{(L)}$, thereby confirming that Proposition 4.2 holds.

## D. Proof of Theorem 4.3

In this section, we provide a detailed proof for Theorem 4.3. We begin by introducing and proving a lemma that demonstrates how $\mathbf{w}_{\text{gd}}^{(0)}$ must align with the direction of $\boldsymbol{\beta}^*$. This alignment constrains the choice of $\mathbf{w}_{\text{gd}}^{(0)}$ to a scalar multiple of $\boldsymbol{\beta}^*$, specifically in the form of $c_0 \cdot \boldsymbol{\beta}^*$. Additionally, in the subsequent sections, we will use the notation $\widehat{\boldsymbol{\Sigma}} = \sum_{i=1}^n \mathbf{x}_i \mathbf{x}_i^\top$.

**Lemma D.1.** *Under the same conditions with Theorem 4.3, to minimize the loss $\mathcal{R}(\mathbf{w}_{\text{gd}}^{(L)})$, $\mathbf{w}_{\text{gd}}^{(0)}$ is always in the form of* $c_0 \cdot \boldsymbol{\beta}^*$.

*Proof of Lemma D.1.* Utilizing the independence among the examples in $T = \{(\mathbf{x}_1, y_1), \ldots, (\mathbf{x}_n, y_n), (\mathbf{x}_{\text{query}}, y_{\text{query}})\}$, and $\mathbf{w}^*$, we can expand $\mathcal{R}_{\mathbf{w}_{\text{gd}}^{(L)}}$ by law of total expectation as

$$
\begin{aligned}
\mathcal{R}(\mathbf{w}_{\text{gd}}^{(L)}) &= \mathbb{E}_{T,\mathbf{w}^*} \left[ \left( \langle \mathbf{w}_{\text{gd}}^{(L)}, \mathbf{x}_{\text{query}} \rangle - \text{sign}(\langle \mathbf{w}^*, \mathbf{x}_{\text{query}} \rangle) \right)^2 \right] \\
&= \mathbb{E}_{T,\mathbf{w}^*} \left[ \langle \mathbf{w}_{\text{gd}}^{(L)}, \mathbf{x}_{\text{query}} \rangle^2 - 2 \text{sign}(\langle \mathbf{w}^*, \mathbf{x}_{\text{query}} \rangle) \langle \mathbf{w}_{\text{gd}}^{(L)}, \mathbf{x}_{\text{query}} \rangle \right] + 1 \\
&= \mathbb{E}_{\{(\mathbf{x}_i, y_i)\}_{i=1}^n, \mathbf{w}^*} \left[ \mathbb{E}_{(\mathbf{x}_{\text{query}}, y_{\text{query}})} \left[ \langle \mathbf{w}_{\text{gd}}^{(L)}, \mathbf{x}_{\text{query}} \rangle^2 - 2 \text{sign}(\langle \mathbf{w}^*, \mathbf{x}_{\text{query}} \rangle) \langle \mathbf{w}_{\text{gd}}^{(L)}, \mathbf{x}_{\text{query}} \rangle \big| \{(\mathbf{x}_i, y_i)\}_{i=1}^n, \mathbf{w}^* \right] \right] + 1 \\
&= \mathbb{E}_{\{(\mathbf{x}_i, y_i)\}_{i=1}^n, \mathbf{w}^*} \left[ \left\| \mathbf{w}_{\text{gd}}^{(L)} - \sqrt{\frac{2}{\pi}} \mathbf{w}^* \right\|_2^2 \right] + 1 - \frac{2}{\pi},
\end{aligned}
$$

where the last equality holds since $\mathbb{E}_{\mathbf{x}_{\text{query}}}[\langle \mathbf{w}, \mathbf{x}_{\text{query}} \rangle^2] = \mathbf{w}^\top \mathbb{E}_{\mathbf{x}_{\text{query}}}[\mathbf{x}_{\text{query}} \mathbf{x}_{\text{query}}^\top] \mathbf{w} = \|\mathbf{w}\|_2^2$ when $\mathbf{w}$ is independent with $\mathbf{x}_{\text{query}}$, and $\mathbb{E}_{\mathbf{x}_{\text{query}}}[\langle \mathbf{w}_1, \mathbf{x}_{\text{query}} \rangle \text{sign}(\langle \mathbf{w}_2, \mathbf{x}_{\text{query}} \rangle)] = \sqrt{\frac{2}{\pi}} \langle \mathbf{w}^*, \mathbf{w}_{\text{gd}}^{(L)} \rangle$ implied by Lemma F.1. Therefore in the next we attempt to optimize the first term $\mathbb{E}_{\{(\mathbf{x}_i, y_i)\}_{i=1}^n, \mathbf{w}^*} \left[ \left\| \mathbf{w}_{\text{gd}}^{(L)} - \sqrt{\frac{2}{\pi}} \mathbf{w}^* \right\|_2^2 \right]$. By applying the closed form of $\mathbf{w}_{\text{gd}}^{(L)}$ in Lemma C.1 with all $\alpha_l = \alpha$, we have

$$
\mathbf{w}_{\text{gd}}^{(L)} = \left( \mathbf{I}_d - \alpha \widehat{\boldsymbol{\Sigma}} \right)^L \cdot \mathbf{w}_{\text{gd}}^{(0)} + \alpha \sum_{l=0}^{L-1} \left( \mathbf{I}_d - \alpha \widehat{\boldsymbol{\Sigma}} \right)^l \cdot \left( \sum_{i=1}^n \text{sign}(\langle \mathbf{w}^*, \mathbf{x}_i \rangle) \mathbf{x}_i \right).
$$

Based on this, we can further derive that

$$
\begin{aligned}
\mathbb{E}_{\{(\mathbf{x}_i, y_i)\}_{i=1}^n, \mathbf{w}^*} \left[ \left\| \mathbf{w}_{\text{gd}}^{(L)} - \sqrt{\frac{2}{\pi}} \mathbf{w}^* \right\|_2^2 \right] =& (\mathbf{w}_{\text{gd}}^{(0)})^\top \mathbb{E}_{\{(\mathbf{x}_i, y_i)\}_{i=1}^n, \mathbf{w}^*} \left[ \left( \mathbf{I}_d - \alpha \widehat{\boldsymbol{\Sigma}} \right)^{2L} \right] \mathbf{w}_{\text{gd}}^{(0)} \\
&- 2\alpha (\mathbf{w}_{\text{gd}}^{(0)})^\top \mathbb{E}_{\{(\mathbf{x}_i, y_i)\}_{i=1}^n, \mathbf{w}^*} \left[ \sum_{l=0}^{L-1} \left( \mathbf{I}_d - \alpha \widehat{\boldsymbol{\Sigma}} \right)^{l+L} \cdot \left( \sum_{i=1}^n \text{sign}(\langle \mathbf{w}^*, \mathbf{x}_i \rangle) \mathbf{x}_i \right) \right] + C \\
=& c_1 \left\| \mathbf{w}_{\text{gd}}^{(0)} \right\|_2^2 - 2c_2 \langle \mathbf{w}_{\text{gd}}^{(0)}, \boldsymbol{\beta}^* \rangle + C \\
=& c_1 \left\| \mathbf{w}_{\text{gd}}^{(0)} - \frac{c_2}{c_1} \boldsymbol{\beta}^* \right\|_2^2 + C - \frac{c_2^2}{c_1}, \quad\quad\quad\quad\quad\quad (\text{D.1})
\end{aligned}
$$

where $c_1, c_2, C$ are some scalar independent of $\mathbf{w}_{\text{gd}}^{(0)}$. The second inequality holds since $\mathbb{E}_{\{(\mathbf{x}_i, y_i)\}_{i=1}^n, \mathbf{w}^*} \big[ \left( \mathbf{I}_d - \alpha \widehat{\boldsymbol{\Sigma}} \right)^{2L} \big] = c_1 \mathbf{I}_d$ for some scalar $c_1$, guaranteed by Lemma F.2, and $\mathbb{E}_{\{(\mathbf{x}_i, y_i)\}_{i=1}^n, \mathbf{w}^*} \big[ \sum_{l=0}^{L-1} \left( \mathbf{I}_d - \alpha \widehat{\boldsymbol{\Sigma}} \right)^{l+L} \cdot \left( \sum_{i=1}^n \text{sign}(\langle \mathbf{w}^*, \mathbf{x}_i \rangle) \mathbf{x}_i \right) \big] = c_2 \boldsymbol{\beta}^*$ for some scalar $c_2$, guaranteed by Lemma F.3. As the result of (C.3) is a quartic function of $\mathbf{w}_{\text{gd}}^{(0)}$, we can easily conclude that it achieves the minimum value when $\mathbf{w}_{\text{gd}}^{(0)} = c_0 \boldsymbol{\beta}^*$ for some scalar $c_0$, which completes the proof. $\qquad\square$

Based on Lemma D.1, in the following proof, we will directly replace $\mathbf{w}_{\mathrm{gd}}^{(0)}$ with $c_0\boldsymbol{\beta}^*$ and attempt to find the optimal $c_0$. Now we are ready to prove the following theorem, a representation of Theorem 4.3.

**Theorem D.2** (Restate of Theorem 4.3). *For training distribution $\mathcal{D}_{\mathrm{tr}}$ in Definition 3.1, suppose that the training context length $n$ is sufficiently large such that $n \geq \widetilde{\Omega}(\max\{d^2, dL\})$. Additionally, suppose that the perturbation of $\mathbf{w}^*$ around its expectation $\boldsymbol{\beta}^*$ is smaller than $\frac{\pi}{2}$, i.e. $\langle \mathbf{w}^*, \boldsymbol{\beta}^* \rangle > 0$. then for any learning rate $\alpha$ and initialization $\mathbf{w}_{\mathrm{gd}}^{(0)}$, it holds that*

$$\mathcal{R}(\mathbf{w}_{\mathrm{gd}}^{(L)}) \leq \Theta\big((1-\alpha n)^L \|\mathbf{w}_{\mathrm{gd}}^{(0)} - c_1\mathbf{w}^*\|_2^2\big) + \widetilde{\Theta}(\alpha dL) + C,$$

*where both $c_1, C$ are absolute constants. Additionally, by taking $\mathbf{w}_{\mathrm{gd}}^{(0)} = c_1\boldsymbol{\beta}^*$ and $\alpha = \widetilde{\Theta}(\frac{1}{nL})$, the upper bound above achieve its optimal rates as*

$$\mathcal{R}(\mathbf{w}_{\mathrm{gd}}^{(L)}) \leq \widetilde{\Theta}\Big(\frac{d}{n}\Big) + C.$$

*Proof of Theorem D.2.* Utilizing the fact that $\mathbf{I}_d - \big(\mathbf{I}_d - \alpha\widehat{\boldsymbol{\Sigma}}\big)^L = \alpha \sum_{l=0}^{L-1} \big(\mathbf{I}_d - \alpha\widehat{\boldsymbol{\Sigma}}\big)^l \widehat{\boldsymbol{\Sigma}}$ and $\mathbf{w}_{\mathrm{gd}}^{(0)} = c_0\boldsymbol{\beta}^*$, we can re-write the close form of $\mathbf{w}_{\mathrm{gd}}^{(L)}$ as

$$\mathbf{w}_{\mathrm{gd}}^{(L)} = \Big(\mathbf{I}_d - (\mathbf{I}_d - \alpha\widehat{\boldsymbol{\Sigma}})^L\Big)\cdot\sqrt{\frac{2}{\pi}}\mathbf{w}^* + \big(\mathbf{I}_d - \alpha\widehat{\boldsymbol{\Sigma}}\big)^L\cdot c_0\boldsymbol{\beta}^* - \alpha\sum_{l=0}^{L-1}\big(\mathbf{I}_d - \alpha\widehat{\boldsymbol{\Sigma}}\big)^l\cdot\Big(\sqrt{\frac{2}{\pi}}\widehat{\boldsymbol{\Sigma}}\mathbf{w}^* - \Big(\sum_{i=1}^n \mathrm{sign}(\langle\mathbf{w}^*, \mathbf{x}_i\rangle)\mathbf{x}_i\Big)\Big)$$

Then by the similar calculation to Lemma D.1, we have

$$\mathcal{R}(\mathbf{w}_{\mathrm{gd}}^{(L)}) = \mathbb{E}\bigg[\Big\|\mathbf{w}_{\mathrm{gd}}^{(L)} - \sqrt{\frac{2}{\pi}}\mathbf{w}^*\Big\|_2^2\bigg] + C$$

$$= \mathbb{E}\bigg[\Big\|(\mathbf{I}_d - \alpha\widehat{\boldsymbol{\Sigma}})^L\cdot\Big(c_0\boldsymbol{\beta}^* - \sqrt{\frac{2}{\pi}}\mathbf{w}^*\Big) - \alpha\sum_{l=0}^{L-1}\big(\mathbf{I}_d - \alpha\widehat{\boldsymbol{\Sigma}}\big)^l\cdot\Big(\sqrt{\frac{2}{\pi}}\widehat{\boldsymbol{\Sigma}}\mathbf{w}^* - \Big(\sum_{i=1}^n \mathrm{sign}(\langle\mathbf{w}^*, \mathbf{x}_i\rangle)\mathbf{x}_i\Big)\Big)\Big\|_2^2\bigg] + C$$

$$\leq 2\underbrace{\mathbb{E}\bigg[\Big\|(\mathbf{I}_d - \alpha\widehat{\boldsymbol{\Sigma}})^L\cdot\Big(c_0\boldsymbol{\beta}^* - \sqrt{\frac{2}{\pi}}\mathbf{w}^*\Big)\Big\|_2^2\bigg]}_{I} + 2\underbrace{\mathbb{E}\bigg[\alpha^2\Big\|\sum_{l=0}^{L-1}\big(\mathbf{I}_d - \alpha\widehat{\boldsymbol{\Sigma}}\big)^l\cdot\Big(\sqrt{\frac{2}{\pi}}\widehat{\boldsymbol{\Sigma}}\mathbf{w}^* - \Big(\sum_{i=1}^n \mathrm{sign}(\langle\mathbf{w}^*, \mathbf{x}_i\rangle)\mathbf{x}_i\Big)\Big)\Big\|_2^2\bigg]}_{II} + C$$

where the last inequality hols by $(a+b)^2 \leq 2a^2 + 2b^2$, and $C$ is an absolute constant. Therefore, in the following, we discuss the upper-bounds for $I$ and $II$ respectively. For $I$, we have

$$I \leq \mathbb{E}\bigg[\Big\|(\mathbf{I}_d - \alpha\widehat{\boldsymbol{\Sigma}})\Big\|_2^{2L}\bigg]\cdot\mathbb{E}\bigg[\Big(c_0\boldsymbol{\beta}^* - \sqrt{\frac{2}{\pi}}\mathbf{w}^*\Big)\Big\|_2^2\bigg] \leq O\big((1-\alpha n)^{2L}\big)\cdot\mathbb{E}\bigg[\Big(c_0\boldsymbol{\beta}^* - \sqrt{\frac{2}{\pi}}\mathbf{w}^*\Big)\Big\|_2^2\bigg],$$

where the first inequality is derived by the independence among $\mathbf{x}_i$ and $\mathbf{w}^*$ and the submultiplicativity of $\ell_2$ norm, and the second inequality holds by the concentration results regarding $\|\widehat{\boldsymbol{\Sigma}}\|_2$ provided in Lemma F.4. For $II$, we can derive that

$$II \leq \underbrace{\mathbb{E}\Big[\alpha^2\sum_{l_1,l_2=0}^{L-1}\big\|\mathbf{I}_d - \alpha\widehat{\boldsymbol{\Sigma}}\big\|_2^{l_1+l_2}\Big]}_{II.1}\underbrace{\mathbb{E}\bigg[\Big\|\sqrt{\frac{2}{\pi}}\widehat{\boldsymbol{\Sigma}}\mathbf{w}^* - \Big(\sum_{i=1}^n \mathrm{sign}(\langle\mathbf{w}^*, \mathbf{x}_i\rangle)\mathbf{x}_i\Big)\Big\|_2^2\bigg]}_{II.2},$$

where the inequality is guaranteed by the submultiplicativity of $\ell_2$ norm. Then we discuss $II.1$ and $II.2$ respectively. For $II.1$, we have

$$II.1 \leq \frac{\alpha}{\|\widehat{\boldsymbol{\Sigma}}\|_2}\sum_{l_1,l_2=0}^{L-1}\frac{1}{l_1+l_2+1} \leq \frac{\alpha L}{\|\widehat{\boldsymbol{\Sigma}}\|_2}\sum_{l_1=0}^{L-1}\frac{1}{l_1+1} \leq O\Big(\frac{\alpha L\log L}{n}\Big).$$

The first inequality holds by the fact that $x(1-x)^k \leq \frac{1}{k+1}$ for $x \in [0,1]$. The second inequality holds by replace $\frac{1}{l_1+l_2+1}$ with its upper bound $\frac{1}{l_1+1}$. The third inequality holds by $\sum_{l_1=0}^{L-1} \frac{1}{l_1+1} \leq \log L$ and $\|\widehat{\mathbf{\Sigma}}\|_2 = \Theta(n)$ demonstrated in Lemma F.4. For $II.2$, we have

$$II.2 = \mathbb{E}\left[\left\|\sqrt{\frac{2}{\pi}}(\widehat{\mathbf{\Sigma}} - n\mathbf{I}_d)\mathbf{w}^* - \left(\sum_{i=1}^n \text{sign}(\langle \mathbf{w}^*, \mathbf{x}_i \rangle)\mathbf{x}_i - n\sqrt{\frac{2}{\pi}}\mathbf{w}^*\right)\right\|_2^2\right]$$

$$\leq \frac{4}{\pi}\mathbb{E}\|\widehat{\mathbf{\Sigma}} - n\mathbf{I}_d\|_2^2 + 2\mathbb{E}\left[\left\|\sum_{i=1}^n \text{sign}(\langle \mathbf{w}^*, \mathbf{x}_i \rangle)\mathbf{x}_i - n\sqrt{\frac{2}{\pi}}\mathbf{w}^*\right\|_2^2\right] \leq \widetilde{O}(nd).$$

The first equality adds and minuses the same term. The first inequality holds by the submultiplicativity of $\ell_2$ norm, and the fact $(a+b)^2 \leq 2a^2 + 2b^2$. The second inequality holds as $\|\widehat{\mathbf{\Sigma}} - n\mathbf{I}_d\|_2 \leq \widetilde{O}(\sqrt{nd})$, proved in Lemma F.4 and $\|\sum_{i=1}^n \text{sign}(\langle \mathbf{w}^*, \mathbf{x}_i \rangle)\mathbf{x}_i - n\sqrt{\frac{2}{\pi}}\mathbf{w}^*\|_2 \leq \widetilde{O}(\sqrt{nd})$, proved in Lemma F.5. Combining all the preceding results, we can obtain that

$$\mathcal{R}(\mathbf{w}_{\text{gd}}^{(L)}) \leq O\left((1-\alpha n)^{2L}\right) \cdot \mathbb{E}\left[\left\|\left(c_0\boldsymbol{\beta}^* - \sqrt{\frac{2}{\pi}}\mathbf{w}^*\right)\right\|_2^2\right] + \widetilde{O}(\alpha dL) + C.$$

It is straightforward that when taking $c_0 = \sqrt{\frac{2}{\pi}}$, the expectation term will achieve its minimum, which is the variance of $\mathbf{w}^*$ multiplying by a factor $\sqrt{\frac{2}{\pi}}$. This finishes the proof that the optimal initialization takes the value as $\mathbf{w}_{\text{gd}}^{(0)} = \sqrt{\frac{2}{\pi}}\boldsymbol{\beta}^*$. We re-plug this result into the upper-bound above and utilize the fact that the variance is at the constant order. Then to find the optimal learning rate $\alpha$ is actually to optimize the summation of $(1-\alpha n)^{2L}$ and $\alpha dL$. We can note that the first term will decrease as $\alpha$ increases, while the second term will increase as $\alpha$ increases. Therefore, minimizing the summation of these two terms is essentially equivalent to finding an optimal $\alpha$ such that both terms are of the same order. Then we can notice that when consider $\alpha = \frac{\log(n/d)}{2nL}$, the first term can be bounded as

$$(1-\alpha n)^{2L} = \left(1 - \frac{\log(n/d)}{2L}\right)^{2L} \leq \frac{d}{n}.$$

Additionally, it is straightforward that $\alpha dL = \frac{d\log(n/d)}{n}$. When omitting the factors of $\log$, we conclude that these two terms are at the same order. Therefore, the optimal choice of learning rate is $\alpha = \widetilde{\Theta}(\frac{1}{nL})$, which can optimize the excess risk as

$$\mathcal{R}(\mathbf{w}_{\text{gd}}^{(L)}) - C \leq \widetilde{O}\left(\frac{d}{n}\right).$$

This completes the proof. $\qquad\square$

Here we provide further discussions regarding the upper bound for the population loss achieved when choosing the optimal learning rate and initialization. The constant $C$ represents an irreducible term arising from the variance of the model. Such an irreducible term always exists when considering least-squares loss, similar to the noise variance in classic linear regression problems. Therefore, when considering the problems with least-square loss function, it is common to define $\mathcal{R}(\mathbf{w}_{\text{gd}}^{(L)}) - C$ as the excess risk and attempt to minimize this term. Consequently, Theorem D.2 reveals that when using the optimal parameters, the excess risk $\mathcal{R}(\mathbf{w}_{\text{gd}}^{(L)}) - C$ will converge to $0$ as the context length $n$ goes to infinity.

## E. Proof of Lemma 4.4 and Theorem 4.5

In this section, we provide the proof for both Lemma 4.4 and Theorem 4.5. W.L.O.G, we assume that $\sigma > 0$ in the subsequent proof. This implies that $y_{\text{hc}} = -1$, $y_{\text{query}} = 1$ and $\mathcal{E}(\mathbf{w}) = \mathbb{P}(\langle \mathbf{w}, \mathbf{x}_{\text{query}} \rangle < 0)$ for any $\mathbf{w}$. Then we first introduce a lemma providing a closed form for $\widetilde{\mathbf{w}}_{\text{gd}}^{(l)}$, which is the parameter vector of the linear model trained by gradient descent with the optimal parameters derived in Theorem 4.3 and data $(\mathbf{x}_{\text{hc}}, y_{\text{hc}})$.

**Lemma E.1.** *For the gradient descent iterates* $\widetilde{\mathbf{w}}_{\text{gd}}^{(l)}$, *it holds that*

$$\widetilde{\mathbf{w}}_{\text{gd}}^{(l)} = c\boldsymbol{\beta}^* + a(l) \cdot \mathbf{x}_\perp \qquad (E.1)$$

*for all $l \in \{0, 1, \ldots, L\}$. $c$ is the coefficient of $\boldsymbol{\beta}^*$ of initialization $\mathbf{w}_{\mathrm{gd}}^{(0)}$ and $a(l)$ follows that*

$$a(l) = -\left(1 - \left(1 - \alpha N\|\mathbf{x}_\perp\|_2^2\right)^l\right)\frac{1 + c\langle\mathbf{x}_\perp, \boldsymbol{\beta}^*\rangle}{\|\mathbf{x}_\perp\|_2^2}$$

*Proof of Lemma E.1.* We prove this lemma by induction. It is straightforward that $\widetilde{\mathbf{w}}_{\mathrm{gd}}^{(0)} = c\boldsymbol{\beta}^*$ and $\widetilde{\mathbf{w}}_{\mathrm{gd}}^{(1)} = c\boldsymbol{\beta}^* - \alpha N\mathbf{x}_\perp$, complying with the formula (E.1). By induction, we assume (E.1) still holds for $l$-th iteration. Then at the $l + 1$-th iteration, we have

$$\begin{aligned}
\widetilde{\mathbf{w}}_{\mathrm{gd}}^{(l+1)} &= \left(\mathbf{I}_d - \alpha N\mathbf{x}_\perp\mathbf{x}_\perp^\top\right) \cdot \widetilde{\mathbf{w}}_{\mathrm{gd}}^{(l)} - \alpha N\mathbf{x}_\perp \\
&= \left(\mathbf{I}_d - \alpha N\mathbf{x}_\perp\mathbf{x}_\perp^\top\right) \cdot \left(c\boldsymbol{\beta}^* + a(l)\mathbf{x}_\perp\right) - \alpha N\mathbf{x}_\perp \\
&= c\boldsymbol{\beta}^* + \left(a(l)(1 - \alpha N\|\mathbf{x}_\perp\|_2^2) - \alpha N(1 + c\langle\mathbf{x}_\perp, \boldsymbol{\beta}^*\rangle)\right) = c\boldsymbol{\beta}^* + a(l+1)\mathbf{x}_\perp.
\end{aligned}$$

Additionally, by the fact $a(l+1) = a(l)(1 - \alpha N\|\mathbf{x}_\perp\|_2^2) - \alpha N(1 + c\langle\mathbf{x}_\perp, \boldsymbol{\beta}^*\rangle)$, we can derive that

$$\begin{aligned}
\left(a(l+1) + \frac{1 + c\langle\mathbf{x}_\perp, \boldsymbol{\beta}^*\rangle}{\|\mathbf{x}_\perp\|_2^2}\right) &= \left(1 - \alpha N\|\mathbf{x}_\perp\|_2^2\right)\left(a(l) + \frac{1 + c\langle\mathbf{x}_\perp, \boldsymbol{\beta}^*\rangle}{\|\mathbf{x}_\perp\|_2^2}\right) \\
&= \cdots \\
&= -\left(1 - \alpha N\|\mathbf{x}_\perp\|_2^2\right)^l \frac{1 + c\langle\mathbf{x}_\perp, \boldsymbol{\beta}^*\rangle}{\|\mathbf{x}_\perp\|_2^2}.
\end{aligned}$$

This implies that

$$a(l) = -\left(1 - \left(1 - \alpha N\|\mathbf{x}_\perp\|_2^2\right)^l\right)\frac{1 + c\langle\mathbf{x}_\perp, \boldsymbol{\beta}^*\rangle}{\|\mathbf{x}_\perp\|_2^2},$$

which completes the proof. □

Based on the closed form of $\widetilde{\mathbf{w}}_{\mathrm{gd}}^{(L)}$ obtained by Lemma E.1, we are ready to prove Lemma 4.4 and Theorem 4.5.

*Proof of Lemma 4.4.* By Lemma E.1, the output of the linear model trained via gradient descent on $\mathbf{x}_{\mathrm{query}}$ can be expanded as

$$\begin{aligned}
\langle\widetilde{\mathbf{w}}_{\mathrm{gd}}^{(L)}, \mathbf{x}_{\mathrm{query}}\rangle &= \langle c\boldsymbol{\beta}^* + a(L)\mathbf{x}_\perp, \mathbf{x}_\perp + \sigma\mathbf{w}^*\rangle \\
&= c\langle\boldsymbol{\beta}^*, \mathbf{x}_\perp\rangle + a(L)\|\mathbf{x}_\perp\|_2^2 + c\sigma\langle\mathbf{w}^*, \boldsymbol{\beta}^*\rangle \\
&= c\langle\boldsymbol{\beta}^*, \mathbf{x}_\perp\rangle - \left(1 - \left(1 - \alpha N\|\mathbf{x}_\perp\|_2^2\right)^L\right)\left(1 + c\langle\mathbf{x}_\perp, \boldsymbol{\beta}^*\rangle\right) + c\sigma\langle\mathbf{w}^*, \boldsymbol{\beta}^*\rangle \\
&= c\sigma\langle\mathbf{w}^*, \boldsymbol{\beta}^*\rangle - 1 + \left(1 - \alpha N\|\mathbf{x}_\perp\|_2^2\right)^L\left(1 + c\langle\mathbf{x}_\perp, \boldsymbol{\beta}^*\rangle\right). \quad (\mathrm{E}.2)
\end{aligned}$$

By utilizing the independence among $\mathbf{w}^*$, $\sigma$, and $\mathbf{x}_\perp$ and law of total expectation, we can derive that

$$\begin{aligned}
\mathcal{E}(\widetilde{\mathbf{w}}_{\mathrm{gd}}^{(L)}) &= \mathbb{P}\left(c\sigma\langle\mathbf{w}^*, \boldsymbol{\beta}^*\rangle - 1 + \left(1 - \alpha N\|\mathbf{x}_\perp\|_2^2\right)^L\left(1 + c\langle\mathbf{x}_\perp, \boldsymbol{\beta}^*\rangle\right) \leq 0\right) \\
&= \mathbb{E}\left[\mathbb{P}\left(c\sigma\langle\mathbf{w}^*, \boldsymbol{\beta}^*\rangle - 1 + \left(1 - \alpha N\|\mathbf{x}_\perp\|_2^2\right)^L\left(1 + c\langle\mathbf{x}_\perp, \boldsymbol{\beta}^*\rangle\right) \leq 0 \Big| \mathbf{w}^*, \mathbf{x}_\perp\right)\right] \\
&= \mathbb{E}\left[F_\sigma\left(\frac{1 - \left(1 - \alpha N\|\mathbf{x}_\perp\|_2^2\right)^L\left(1 + c\langle\mathbf{x}_\perp, \boldsymbol{\beta}^*\rangle\right)}{c\langle\mathbf{w}^*, \boldsymbol{\beta}^*\rangle}\right)\right], \quad (\mathrm{E}.3)
\end{aligned}$$

where $F_\sigma(\cdot)$ is the cumulative distribution function of $\sigma$. Similarly, we also have

$$\mathcal{E}(\mathbf{w}_{\mathrm{gd}}^{(0)}) = \mathbb{E}\left[F_\sigma\left(-\frac{\langle\mathbf{x}_\perp, \boldsymbol{\beta}^*\rangle}{\langle\mathbf{w}^*, \boldsymbol{\beta}^*\rangle}\right)\right].$$

Therefore, by Taylor's first order expansion, we have

$$\mathcal{E}(\widetilde{\mathbf{w}}_{\text{gd}}^{(L)}) - \mathcal{E}(\mathbf{w}_{\text{gd}}^{(0)}) = \mathbb{E}\left[F'_\sigma(\boldsymbol{\xi})\frac{\left(1 - (1 - \alpha N\|\mathbf{x}_\perp\|_2^2)^L\right)\left(1 + c\langle\mathbf{x}_\perp, \boldsymbol{\beta}^*\rangle\right)}{1 + c\langle\mathbf{w}^*, \boldsymbol{\beta}^*\rangle}\right]$$

$$\leq \left(1 - \left(1 - \Theta\left(\frac{Nd}{nL}\right)\right)^L\right)\widetilde{O}(\sqrt{d}) \leq \widetilde{O}\left(\frac{Nd^{3/2}}{n}\right) \leq \widetilde{o}(1),$$

where the first inequality utilizing the concentration results that $\|\mathbf{x}_\perp\|_2^2 \Theta(d)$ and $\langle\mathbf{x}_\perp, \boldsymbol{\beta}^*\rangle = \widetilde{O}(\sqrt{d})$. The second inequality holds by the fact $\left(1 - \Theta\left(\frac{Nd}{nL}\right)\right)^L = 1 - \Theta\left(\frac{Nd}{n}\right)$ by our condition $n \leq o(d^{3/2}/n)$, which also implies the last inequality holds. Therefore, we finish the proof. □

In the next, we prove Theorem 4.5

*Proof of Theorem 4.5.* Similar to the proof of Lemma 4.4, we have that

$$\mathcal{E}(\widetilde{\mathbf{w}}_{\text{gd}}^{(L)}) = \mathbb{E}\left[F_\sigma\left(\frac{1 - (1 - \alpha N\|\mathbf{x}_\perp\|_2^2)^L\left(1 + c\langle\mathbf{x}_\perp, \boldsymbol{\beta}^*\rangle\right)}{c\langle\mathbf{w}^*, \boldsymbol{\beta}^*\rangle}\right)\right]$$

$$= \mathbb{E}\left[F_\sigma\left(\frac{1 - (1 - \alpha N\|\mathbf{x}_\perp\|_2^2)^L}{c\langle\mathbf{w}^*, \boldsymbol{\beta}^*\rangle}\right) - F'_\sigma(\boldsymbol{\xi})\frac{(1 - \alpha N\|\mathbf{x}_\perp\|_2^2)^L|\langle\mathbf{x}_\perp, \boldsymbol{\beta}^*\rangle|\operatorname{sign}(\langle\mathbf{x}_\perp, \boldsymbol{\beta}^*\rangle)}{\langle\mathbf{w}^*, \boldsymbol{\beta}^*\rangle}\right]$$

$$= \mathbb{E}\left[F_\sigma\left(\frac{1 - (1 - \alpha N\|\mathbf{x}_\perp\|_2^2)^L}{c\langle\mathbf{w}^*, \boldsymbol{\beta}^*\rangle}\right)\right] = \mathbb{E}\left[F_\sigma\left(\frac{1 - (1 - \widetilde{\Theta}(\frac{Nd}{nL}))^L}{c\langle\mathbf{w}^*, \boldsymbol{\beta}^*\rangle}\right)\right],$$

where the third inequality holds as $\operatorname{sign}(\langle\mathbf{x}_\perp, \boldsymbol{\beta}^*\rangle)$ is independent with $|\langle\mathbf{x}_\perp, \boldsymbol{\beta}^*\rangle|$ and $\|\mathbf{x}_\perp\|_2^2$, and $F'(\boldsymbol{\xi})$ is a constant. Additionally, let $\sigma$ follows the uniform distribution from $a$ to $b$, then we can expand the expectation above as

$$\mathcal{E}(\widetilde{\mathbf{w}}_{\text{gd}}^{(L)}) = \mathbb{E}\left[F_\sigma\left(\frac{1 - (1 - \widetilde{\Theta}(\frac{Nd}{nL}))^L}{c\langle\mathbf{w}^*, \boldsymbol{\beta}^*\rangle}\right)\mathbb{1}\left\{\frac{1 - (1 - \widetilde{\Theta}(\frac{Nd}{nL}))^L}{c\langle\mathbf{w}^*, \boldsymbol{\beta}^*\rangle} \leq a\right\}\right]$$

$$+ \mathbb{E}\left[F_\sigma\left(\frac{1 - (1 - \widetilde{\Theta}(\frac{Nd}{nL}))^L}{c\langle\mathbf{w}^*, \boldsymbol{\beta}^*\rangle}\right)\mathbb{1}\left\{a < \frac{1 - (1 - \widetilde{\Theta}(\frac{Nd}{nL}))^L}{c\langle\mathbf{w}^*, \boldsymbol{\beta}^*\rangle} \leq b\right\}\right]$$

$$+ \mathbb{E}\left[F_\sigma\left(\frac{1 - (1 - \widetilde{\Theta}(\frac{Nd}{nL}))^L}{c\langle\mathbf{w}^*, \boldsymbol{\beta}^*\rangle}\right)\mathbb{1}\left\{\frac{1 - (1 - \widetilde{\Theta}(\frac{Nd}{nL}))^L}{c\langle\mathbf{w}^*, \boldsymbol{\beta}^*\rangle} > b\right\}\right]$$

$$= \left(1 - \left(1 - \widetilde{\Theta}\left(\frac{Nd}{nL}\right)\right)^L\right)\mathbb{E}\left[\frac{1}{c\langle\mathbf{w}^*, \boldsymbol{\beta}^*\rangle}\mathbb{1}\left\{a < \frac{1 - (1 - \widetilde{\Theta}(\frac{Nd}{nL}))^L}{c\langle\mathbf{w}^*, \boldsymbol{\beta}^*\rangle} \leq b\right\}\right] + \mathbb{E}\left[\mathbb{1}\left\{\frac{1 - (1 - \Theta(\frac{Nd}{nL}))^L}{c\langle\mathbf{w}^*, \boldsymbol{\beta}^*\rangle} > b\right\}\right]$$

$$= c_1 - c_2\left(1 - \widetilde{\Theta}\left(\frac{Nd}{nL}\right)\right)^L$$

where $c_1, c_2$ are two positive scalars solely depending on $a, b$ and the distribution of $\mathbf{w}^*$. This completes the proof. □

# F. Technical lemmas

In this section, we introduce and prove some technical lemmas utilized in the previous proof.

**Lemma F.1.** *Let* $\mathbf{x} \sim \mathcal{N}(\mathbf{0}_d, \mathbf{I}_d)$, *and* $\mathbf{w}_1, \mathbf{w} \in \mathbb{R}^d$ *be two vectors independent of* $\mathbf{x}$, *with* $\|\mathbf{w}_1\|_2 = 1$, *then it holds that*

$$\mathbb{E}_\mathbf{x}[\langle\mathbf{w}, \mathbf{x}\rangle\operatorname{sign}(\langle\mathbf{w}_1, \mathbf{x}\rangle)] = \sqrt{\frac{2}{\pi}}\langle\mathbf{w}, \mathbf{w}_1\rangle.$$

*Proof of Lemma F.1.* Since $\|\mathbf{w}_1\|_2 = 1$, let $\boldsymbol{\Gamma} = [\mathbf{w}_1, \mathbf{w}_2, \ldots, \mathbf{w}_d] \in \mathbb{R}^d$ be the orthogonal matrix with $\mathbf{w}_1$ being its first column. Then we have

$$\mathbb{E}_{\mathbf{x}}[\langle \mathbf{w}, \mathbf{x} \rangle \operatorname{sign}(\langle \mathbf{w}_1, \mathbf{x} \rangle)] = \mathbb{E}_{\mathbf{x}}[\mathbf{w}^\top \boldsymbol{\Gamma}\boldsymbol{\Gamma}^\top \mathbf{x} \operatorname{sign}(\langle \mathbf{w}_1, \mathbf{x} \rangle)]$$

$$= \sum_{k=1}^{d} \langle \mathbf{w}, \mathbf{w}_k \rangle \mathbb{E}_{\mathbf{x}}[\langle \mathbf{w}_k, \mathbf{x} \rangle \operatorname{sign}(\langle \mathbf{w}_1, \mathbf{x} \rangle)] = \sqrt{\frac{2}{\pi}} \langle \mathbf{w}, \mathbf{w}_1 \rangle,$$

where the last equality holds since $\langle \mathbf{w}_k, \mathbf{x} \rangle \sim \mathcal{N}(0, 1)$ for all $k \in [d]$, $\langle \mathbf{w}_{k_1}, \mathbf{x} \rangle$ and $\langle \mathbf{w}_{k_2}, \mathbf{x} \rangle$ are independent when $k_1 \neq k_2$, and $\mathbb{E}[\langle \mathbf{w}_1, \mathbf{x} \rangle \operatorname{sign}(\langle \mathbf{w}_1, \mathbf{x} \rangle)] = \mathbb{E}[|\langle \mathbf{w}_1, \mathbf{x} \rangle|] = \sqrt{\frac{2}{\pi}}$. This completes the proof. $\square$

For the next lemmas, we follow the notation we used in previous section that $\widehat{\boldsymbol{\Sigma}} = \sum_{i=1}^{n} \mathbf{x}_i \mathbf{x}_i^\top$.

**Lemma F.2.** *For any $k \in \mathbb{N}$, it holds that $\mathbb{E}[\widehat{\boldsymbol{\Sigma}}^k] = c\mathbf{I}_d$, where $c$ is a scalar.*

*Proof of Lemma F.2.* Let $\boldsymbol{\Gamma}$ be any orthogonal matrix, then we have $\boldsymbol{\Gamma}\mathbf{x}_i \sim \mathcal{N}(\mathbf{0}_d, \mathbf{I}_d)$. This implies that $\sum_{i=1}^{n} (\boldsymbol{\Gamma}\mathbf{x}_i)(\boldsymbol{\Gamma}\mathbf{x}_i)^\top$ has the same distribution with $\widehat{\boldsymbol{\Sigma}}$. Therefore, we can derive that

$$\boldsymbol{\Gamma}\mathbb{E}[\widehat{\boldsymbol{\Sigma}}^k]\boldsymbol{\Gamma} = \mathbb{E}\Big[\Big( \sum_{i=1}^{n} (\boldsymbol{\Gamma}\mathbf{x}_i)(\boldsymbol{\Gamma}\mathbf{x}_i)^\top \Big)^k\Big] = \mathbb{E}[\widehat{\boldsymbol{\Sigma}}^k]$$

holds for any orthogonal matrix $\boldsymbol{\Gamma}$, which implies that $\mathbb{E}[\widehat{\boldsymbol{\Sigma}}^k]$ must be at the form $c\mathbf{I}_d$. This completes the proof. $\square$

Lemma F.2 implies that $\mathbb{E}[(\mathbf{I}_d - \widehat{\boldsymbol{\Sigma}})^k] = c\mathbf{I}_d$ for some scalar $c$ as by binomial formula it can be expanded as a summation of polynomials of $\widehat{\boldsymbol{\Sigma}}$, which all have the expectations with the form $c\mathbf{I}_d$.

**Lemma F.3.** *For any $k \in \mathbb{N}$, it holds that*

$$\mathbb{E}\Big[\widehat{\boldsymbol{\Sigma}}^k \Big( \sum_{i=1}^{n} \mathbf{x}_i y_i \Big)\Big] = c\boldsymbol{\beta}^*,$$

*where $c$ is some scalar.*

*Proof of Lemma F.3.* By binomial theorem, we have

$$\mathbb{E}\Big[\widehat{\boldsymbol{\Sigma}}^k \Big( \sum_{i=1}^{n} \mathbf{x}_i y_i \Big)\Big] = \sum_{i=1}^{n} \sum_{k_1=0}^{k} \binom{k}{k_1} \mathbb{E}\Big[\Big( \sum_{i' \neq i} \mathbf{x}_{i'}\mathbf{x}_{i'}^\top \Big)^{k-k_1}\Big] \mathbb{E}[(\mathbf{x}_i\mathbf{x}_i^\top)^{k_1} \mathbf{x}_i y_i].$$

By Lemma F.2, we already obtain that $\mathbb{E}\Big[\Big( \sum_{i' \neq i} \mathbf{x}_{i'}\mathbf{x}_{i'}^\top \Big)^{k-k_1}\Big] = c\mathbf{I}_d$ for some scalar $c$. In the next, it suffices to show that $\mathbb{E}[(\mathbf{x}_i\mathbf{x}_i^\top)^{k_1} \mathbf{x}_i y_i] = c\boldsymbol{\beta}^*$ for some scalar $c$. Since $\|\mathbf{w}^*\|_2 = 1$, let $\boldsymbol{\Gamma} = [\mathbf{w}^*, \mathbf{w}_2, \ldots, \mathbf{w}_d] \in \mathbb{R}^d$ be the orthogonal matrix with $\mathbf{w}^*$ being its first column, and let $\mathbf{x}_i' = \boldsymbol{\Gamma}^\top \mathbf{x}_i \sim \mathcal{N}(0, \mathbf{I}_d)$. This implies that $y_i = \operatorname{sign}(\langle \mathbf{w}^*, \mathbf{x}_i \rangle) = \operatorname{sign}(\mathbf{x}_{i,1}')$, which is the first coordinate of $\mathbf{x}_i'$. Based on this, for any fixed $\mathbf{w}^*$, we can further derive that

$$\mathbb{E}[(\mathbf{x}_i\mathbf{x}_i^\top)^{k_1} \mathbf{x}_i y_i | \mathbf{w}^*] = \boldsymbol{\Gamma}\mathbb{E}[(\mathbf{x}_i'\mathbf{x}_i'^\top)^{k_1} \mathbf{x}_i' \operatorname{sign}(\mathbf{x}_{i,1}')] = \boldsymbol{\Gamma}\mathbb{E}[\|\mathbf{x}_i'\|_2^{2k_1} \mathbf{x}_i' \operatorname{sign}(\mathbf{x}_{i,1}')] = c\mathbf{w}^*.$$

The last equality holds as $\|\mathbf{x}_i'\|_2^{2k_1}$ is a even function for each coordinate of $\mathbf{x}_i'$, which implies that $\mathbb{E}[\|\mathbf{x}_i'\|_2^{2k_1} \mathbf{x}_{i,j}' \operatorname{sign}(\mathbf{x}_{i,1}')] = 0$ for any $j \in [d]$ and $j \neq 1$. Therefore, we can finally obtain that

$$\mathbb{E}[(\mathbf{x}_i\mathbf{x}_i^\top)^{k_1} \mathbf{x}_i y_i] = \mathbb{E}\big[\mathbb{E}[(\mathbf{x}_i\mathbf{x}_i^\top)^{k_1} \mathbf{x}_i y_i | \mathbf{w}^*]\big] = c\mathbb{E}[\mathbf{w}^*] = c\boldsymbol{\beta}^*,$$

which completes the proof. $\square$

**Lemma F.4** (Theorem 9 in Bartlett et al. (2020)). *For any $\delta > 0$, with probability at least $1 - \delta$, it holds that,*

$$\Big\| \frac{1}{n}\widehat{\boldsymbol{\Sigma}} - \mathbf{I}_d \Big\|_2 \leq O\bigg( \max\bigg\{ \frac{d}{n}, \sqrt{\frac{d}{n}}, \frac{\log(1/\delta)}{n}, \sqrt{\frac{\log(1/\delta)}{n}} \bigg\} \bigg).$$

**Lemma F.5.** *For any $\delta > 0$, with probability at least $1 - \delta$, it holds that,*

$$\left\| \sum_{i=1}^{n} \text{sign}(\langle \mathbf{w}^*, \mathbf{x}_i \rangle) \mathbf{x}_i - n\sqrt{\frac{2}{\pi}} \mathbf{w}^* \right\|_2 \le O\left(\sqrt{nd \log(d/\delta)}\right).$$

*Proof of Lemma F.5.* Similar to the previous proof technique, let $\boldsymbol{\Gamma} = [\mathbf{w}^*, \mathbf{w}_2, \ldots, \mathbf{w}_d] \in \mathbb{R}^d$ be the orthogonal matrix with $\mathbf{w}^*$ being its first column, and let $\mathbf{x}'_i = \boldsymbol{\Gamma}^\top \mathbf{x}_i \sim \mathcal{N}(0, \mathbf{I}_d)$. Then we can derive that

$$\sum_{i=1}^{n} \text{sign}(\langle \mathbf{w}^*, \mathbf{x}_i \rangle) \mathbf{x}_i - n\sqrt{\frac{2}{\pi}} \mathbf{w}^* = \left[ \sum_{i=1}^{n} \left( |\mathbf{x}'_{i,1}| - \sqrt{\frac{2}{\pi}} \right) \right] \cdot \mathbf{w}^* + \sum_{j=2}^{d} \left[ \sum_{i=1}^{n} \text{sign}(\mathbf{x}'_{i,1}) \mathbf{x}'_{i,j} \right] \cdot \mathbf{w}_j.$$

Since $|\mathbf{x}'_{i,1}|$ is a subgaussian random variable with expectation $\sqrt{\frac{2}{\pi}}$, by Hoeffding's inequality we can derive that with probability at least $1 - \delta/d$,

$$\sum_{i=1}^{n} \left( |\mathbf{x}'_{i,1}| - \sqrt{\frac{2}{\pi}} \right) \le O\left(\sqrt{n \log(d/\delta)}\right).$$

Additionally, when $j \ne 1$, $\text{sign}(\mathbf{x}'_{i,1}) \mathbf{x}'_{i,j}$ still follows a standard normal distribution (A standard normal random variable times an independent Rademacher random variable is still a standard normal random). Therefore, we can also derive that

$$\sum_{i=1}^{n} \text{sign}(\mathbf{x}'_{i,1}) \mathbf{x}'_{i,j} \le O\left(\sqrt{n \log(d/\delta)}\right)$$

holds with probability at least $1 - \frac{\delta}{d}$. Then by taking an union bound, we can finally obtain that

$$\left\| \sum_{i=1}^{n} \text{sign}(\langle \mathbf{w}^*, \mathbf{x}_i \rangle) \mathbf{x}_i - n\sqrt{\frac{2}{\pi}} \mathbf{w}^* \right\|_2^2 = \left[ \sum_{i=1}^{n} \left( |\mathbf{x}'_{i,1}| - \sqrt{\frac{2}{\pi}} \right) \right]^2 + \sum_{j=2}^{d} \left[ \sum_{i=1}^{n} \text{sign}(\mathbf{x}'_{i,1}) \mathbf{x}'_{i,j} \right]^2 \le O\left(nd \log(d/\delta)\right).$$

The first equality holds by the orthogonality among $\mathbf{w}^*, \mathbf{w}_2, \ldots, \mathbf{w}^*$. This completes the proof. $\qquad \square$

**Lemma F.6.** *For any $\delta > 0$, with probability at least $1 - \delta$, it holds that,*

$$\left| \|\mathbf{x}_\perp\|_2^2 - (d-1) \right| \le O\left(\sqrt{d \log(1/\delta)}\right);$$
$$\left| \langle \mathbf{x}_\perp, \boldsymbol{\beta}^* \rangle \right| \le O\left(\sqrt{d \log(1/\delta)}\right).$$

*Proof of Lemma F.6.* By the fact that $\|\mathbf{x}_\perp\|_2^2 \sim \chi_{d-1}^2$, we have $\mathbb{E}[\|\mathbf{x}_\perp\|_2^2] = d - 1$. Then by the Bernstein's inequality, we can obtain that

$$\left| \|\mathbf{x}_\perp\|_2^2 - (d-1) \right| \le O\left(\sqrt{d \log(1/\delta)}\right)$$

holds with probability at least $1 - \delta/2$. Besides, since $\langle \mathbf{x}_\perp, \boldsymbol{\beta}^* \rangle \sim \mathcal{N}(0, 1 - \langle \mathbf{w}^*, \boldsymbol{\beta}^* \rangle)$, by applying the tail bounds of Gaussian distribution, we can obtain that

$$\left| \langle \mathbf{x}_\perp, \boldsymbol{\beta}^* \rangle \right| \le O\left(\sqrt{d \log(1/\delta)}\right)$$

holds with probability at least $1 - \delta/2$. By applying a union bound, we obtain the final result. $\qquad \square$

# G. Experimental setup

## G.1. Context hijacking in LLMs

This section will describe our experimental setup for context hijacking on LLMs of different depths. We first construct four datasets for different tasks, including language, country, sports, and city. The samples in each dataset consist of four parts: prepend, error result, query, and correct result. Each task has a fixed template for the sample. For the language,

the template is "{People} did not speak {error result}.  The native language of {People} is {correct result}". For the country, the template is "{People} does not live in {error result}. {People} has a citizenship from {correct result}". For the sports, the template is "{People} is not good at playing {error result}.  {People}'s best sport is {correct result}". For the city, the template is "{Landmarks} is not in {error result}.  {Landmarks} is in the city of {correct result}". We allow samples to have certain deviations from the templates, but they must generally conform to the semantics of the templates. Instance always match the reality, and the main source of instances is the *CounterFact* dataset (Meng et al., 2022). In our dataset, each task contains three hundred to seven hundred specific instances. We conduct experiments on GPT2 (Radford et al., 2019) of different sizes. Specifically, we consider GPT2, GPT2-MEDIUM, GPT2-LARGE, and GPT2-XL. They have 12 layers, 24 layers, 36 layers, and 48 layers, respectively. We construct a pipeline that test each model on each task, recording the number of prepends for which the context just succeeded in perturbing the output. For those samples that fail to perturb within a certain number of prepends (which is determined by the maximum length of the pre-trained model), we exclude them from the statistics. Finally, we verify the relationship between model depth and robustness by averaging the number of prepends required to successfully perturb the output.

### G.2. Numerical experiments

We use extensive numerical experiments to verify our theoretical results, including gradient descent and linear transformers.

**Gradient descent:** We use a single-layer neural network as the gradient descent model, which contains only one linear hidden layer. Its input dimension is the dimension $d$ of feature $\mathbf{x}$, and we mainly experiment on $d = \{15, 20, 25\}$. Its output dimension is 1, because we only need to judge the classification result by its sign. We use the mean square error as the loss function and *SGD* as the optimizer. All data comes from the defined training distribution $\mathcal{D}_{\mathrm{tr}}$. The hyperparameters we set include training context length $N = 50$, mean of the Gaussian distribution $\boldsymbol{\beta}^\star = 1$, variance of the Gaussian distribution $\boldsymbol{\Sigma} = 0.1$ (then normalized). We initialize the neural network to $c\boldsymbol{\beta}$, and then perform gradient descents with steps $Steps = \{1, 2, ..., 8\}$ and learning rate $lr$. We use grid search to search for the optimal $c$ and $lr$ for the loss function. This is equivalent to the trained transformers of layers 1 to 8 learning to obtain the shared signal $c\boldsymbol{\beta}$ and the optimal learning rate $lr$ for the corresponding number of layers. Then they can use in-context data to fine-tune $c\boldsymbol{\beta}$ to a specific $\mathbf{w}^\star$.

After obtaining the optimal initialization and learning rate, we test it on the dataset from $\mathcal{D}_{\mathrm{te}}$. Again, we set exactly the same hyperparameters as above. In addition, we set the $\sigma$ in the test distribution to 0.1.

**Linear Transformer:** We train on multi-layer single-head linear transformers and use *Adam* as the optimizer. The training settings for models with different numbers of layers are exactly the same. We use the initial learning rate $lr \in \{0.0001, 0.0002\}$, and the training steps are 600,000. We use a learning rate decay mechanism, where the learning rate is decayed by half every 50,000 training steps. For training and testing data, we set the data dimension $d = 20$ and the training context length $N = \{10, 20, 30, 40\}$. We use a batchsize of 5,000 and apply gradient clipping with a threshold of 1.

## H. Additional experiments

### H.1. Robustness of standard transformers with different number of layers

To generalize the results to more realistic settings, we transfer the experiments from linear transformers to larger and standard transformers, such as GPT-2 (Radford et al., 2019). We train and test GPT-2 with different numbers of layers based on exactly the same settings as the linear transformers experiments. The results once again verify our theory (Figure 5). As the context length increases, the model's accuracy decreases, but increasing the number of layers of the model significantly improves the robustness, indicating that our theory has more practical significance. Then we describe the setup of standard transformers experiments briefly.

**Setup:** We use the standard transformers of the GPT-2 architecture for the experiments, and the main settings are similar to (Garg et al., 2022). We set the embedding size to 256, the number of heads to 8, and the batch size to 64. We use a learning rate decay mechanism similar to linear transformers experiments, with an initial learning rate of 0.0002, and then reduced by half every 200,000 steps, for a total of 600,000 steps. We use *Adam* as the optimizer.

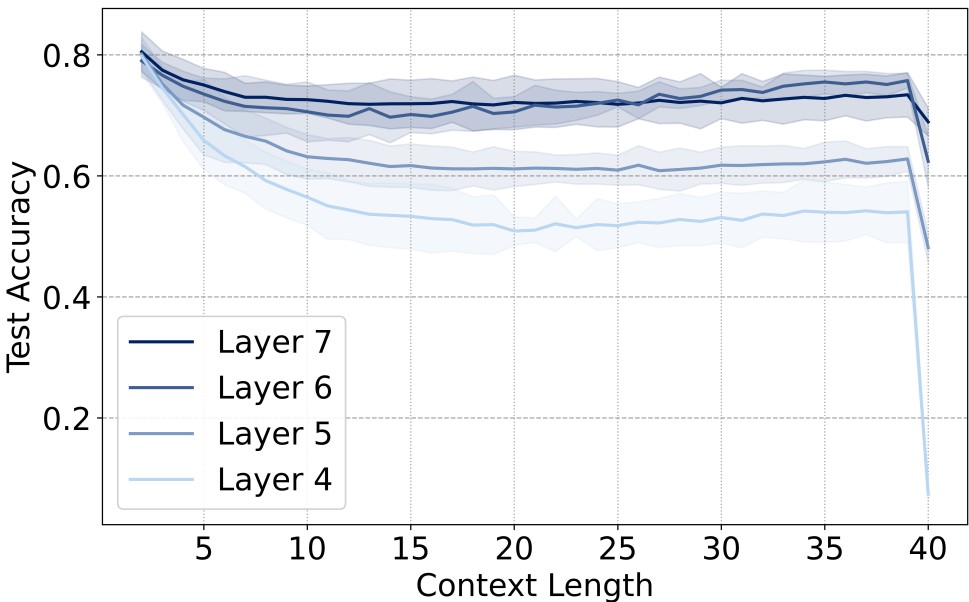

*Figure 5.* Standard transformers experiments with different depths. Testing the trained standard transformers (GPT-2 architecture (Radford et al., 2019)) on the test set, as the number of interference samples increases, the model classification accuracy decreases and gradually converges. The results also show that deeper models are more robust.

### H.2. Linear transformers facing different interference intensity

In this section, we mainly discuss how the robustness of the model changes with the interference intensity. In our modeling, the interference intensity is determined only by the distance between the query sample and the similar interference samples defined in the test set, that is, by the variable $\sigma$ in $\mathcal{D}_{\text{te}}$. In real-world observations, according to the idea of the induction head (Olsson et al., 2022), the more similar the context prepend used for interference is to the query, the more likely the model is to use in-context learning to output incorrect results. Therefore, we examine different $\sigma$ to determine whether the model conforms to the actual real-world interference situation, that is, to verify the rationality of our modeling.

Observing the experiment results in Figure 6, when $\sigma$ gradually decreases from 0.8 to 0.1, that is, the interference intensity of the data gradually increases, the classification accuracy of the model decreases significantly. When $\sigma$ is larger and the interference context is less, the model can always classify accurately, indicating that weak interference does not affect the performance of the model, which is consistent with real observations. Various experimental phenomena show that our modeling of the context hijacking task by the distance between the interference sample and the query sample is consistent with the real semantics.

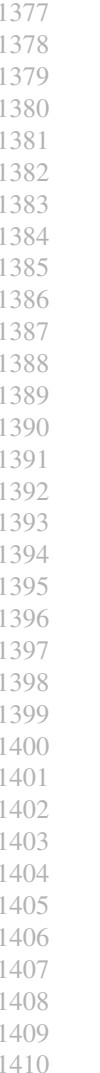
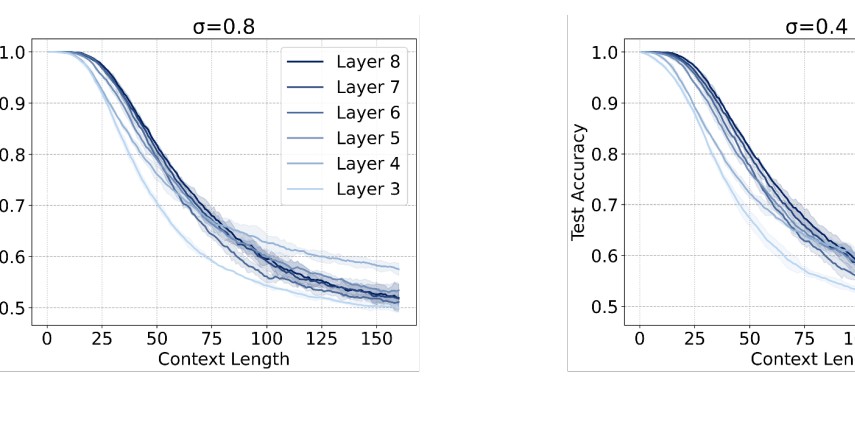
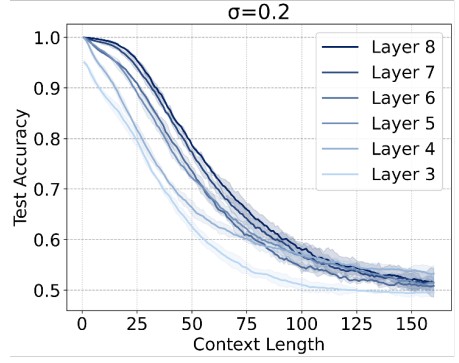
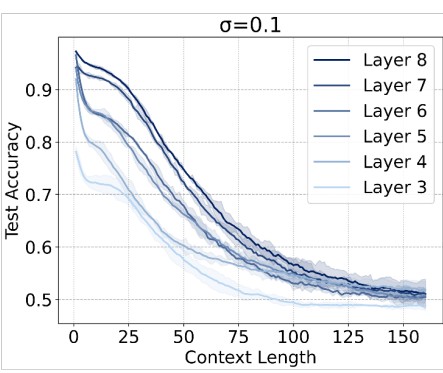

*Figure 6.* Linear transformers experiments with different depths and different $\sigma$. In real-world semantics, smaller $\sigma$ means stronger interference. Comparing the test performance of the model under different $\sigma$, we can find that as $\sigma$ decreases, the robustness of the model decreases significantly, which verifies the rationality of our modeling.

