# OpenReview forum: "On the Robustness of Transformers against Context Hijacking for Linear Classification"
_ICML.cc/2025/Conference — Submitted to ICML 2025_

### Official Review · Reviewer_enhE · 2025-03-11

**Overall Recommendation:** 3

**Summary:**

This paper studies the robustness of Transformers against context hijacking in a linear classification setting.
Empirically, the paper observes deeper transformer can achieve higher robustness. Theoretically, the paper explains this phenomenon as deeper model corresponding to more fine-grained optimization steps, which improves context hijacking robustness.

## update after rebuttal

I would like to keep my original score. On the positive side, I appreciate the paper’s solid analysis and clear exposition. However, the analysis is restricted to the representation level, which is relatively well-established in the literature on ICL. Given that there are already works studying the learnability of ICL, the lack of results on this aspect limits the contribution in my view and is a key reason why I do not increase my score.

**Claims And Evidence:**

Yes.

**Essential References Not Discussed:**

No.

**Experimental Designs Or Analyses:**

Yes.

**Methods And Evaluation Criteria:**

Yes.

**Other Comments Or Suggestions:**

N/A

**Other Strengths And Weaknesses:**

Strengths:
The paper studies the robustness of context hijacking under a well-formulated ICL linear classification setting. Under the setting, the paper provides good theoretical analysis, which is consistent with empirical observations about the relations between the robustness, the training context length, the number of hijacked context examples, and the depth of the transformer model.

Weaknesses:
1. It is unclear how the design of hijacked context data in the linear classification setting is connected to the context hijacking in practice. While the authors claim this "follows the general design of many previous theoretical works", a brief illustration might be essential.

2. The theoretical analysis is restricted to the representation level. It only shows that there exist Transformers that satisfy the desired properties (i.e., equivalence between $L$-layer transformers and $L$ steps gradient descent), based on which the results about the context hijacking robustness can be derived. There lacks analysis on the learnability that the model will necessarily learn the desired paramters from the training data.

**Questions For Authors:**

1. Can you explain how the design of hijacked context data in the linear classification setting is connected to the context hijacking in practice?

2. Though strict theoretical analysis might be beyond the scope of this work, can you provide any explanation or insight about the learnability issue?

**Relation To Broader Scientific Literature:**

N/A

**Theoretical Claims:**

Yes.

---

> ### Author Rebuttal · Authors · 2025-04-01
>
> We appreciate your constructive questions and suggestions! We address them as follows:
>
> >**Q1:** More clarity on the design of hijacked context data in the linear classification setting, and its connection to context hijacking in practice.
>
> **A1:**:
> We carefully design the context hijacking sample $\mathbf{Z}\_{\mathrm{hc}}$ to simulate the context hijacking phenomenon. We will explain it and its connection with context hijacking in practice with the example shown in Figure 1. Our data consists of $\mathbf{x}\_{\mathrm{hc}}, y\_{\mathrm{hc}}, \mathbf{x}\_{\mathrm{query}}$ and $y\_{\mathrm{query}}$. We can assume that $\mathbf{x}\_{\mathrm{hc}}$ = “Rafael Nadal is not good at playing”, $y\_{\mathrm{hc}}$ = “basketball”, $ \mathbf{x}\_{\mathrm{query}}$ = "Rafael Nadal’s best sport is", and $y\_{\mathrm{query}}$ = "tennis".
>
> So our data structure is defined as follows. It consists of two parts: context and query. For each sample $\mathbf{Z}\_{\mathrm{hc}}$, each of its first $n$ columns of context is exactly the same, that is, it is composed of $n$ repeated $(\mathbf{x}\_{\mathrm{hc}},y\_{\mathrm{hc}})$. However, the last query is $(\mathbf{x}\_{\mathrm{query}},y\_{\mathrm{query}})$, which is not equal to $(\mathbf{x}\_{\mathrm{hc}},y\_{\mathrm{hc}})$. This corresponds to the repeated context sentence "Rafael Nadal is not good at playing basketball" and the final query “Rafael Nadal’s best sport is” in practical case.
>
> We design $\mathbf{x}\_{\mathrm{hc}}$ and $\mathbf{x}\_{\mathrm{query}}$ to be different, which satisfies that corresponding $y\_{\mathrm{hc}}$ and $y\_{\mathrm{query}}$ are opposite. We can see that $\mathbf{Z}\_{\mathrm{hc}}$ is designed to perfectly fit the context hijacking phenomenon, consisting of two parts.
>
> * $N$ repeated context samples correspond to repeated interference samples in the context hijacking phenomenon.
> * The values of $\mathbf{x}\_{\mathrm{query}}$ and $\mathbf{x}\_{\mathrm{hc}}$ could be close (controlled by $\sigma$), which is consistent with the close semantics of context in the real context hijacking phenomenon. And the labels of the context samples are opposite to the final predicted label, aligning with the practical observation that context hijacking causes the prediction to flip to the token in the context.
>
> >**Q2:** Theoretical analysis is restricted to the representation level. Any insights about the learnability issue?
>
> **A2**:
> We recognize that the optimization analysis and learnability of multi-layer transformers are of great interest and importance. However, to our knowledge, current research primarily focuses on shallow architectures (one or two layers) [1, 2, 3, 4]. At this stage, it appears to be exceedingly challenging and nearly impossible to offer a rigorous theoretical analysis of the optimization processes for multi-layer transformers.
>
> While lacking rigorous derivation, we have some plausible hypotheses regarding the training dynamics for multi-layer. Proposition 4.2 suggests that the choice of learning rates is symmetric across all steps. We conjecture that this conclusion extends to the context of global optimization, implying that meaning our Proposition 4.2 provide insightful implications from the perspective of training. This is supported by Theorem 2.1 in [5], which shows that the difference in the $\ell$-2 norm across layers in a deep homogeneous neural network remains constant during training. Such a conclusion is applicable to linear transformers, as they are always homogeneous. Therefore, if all layers of linear transformers are initialized from the same point, they will behave similarly throughout the training process. Based on this conjecture, the matrix factorization technique proposed in [2] for one-layer linear transformers might also be applicable to general multi-layer transformers. We believe this is an interesting and promising future work direction.
>
> [1] Zhang, et al. "In-context learning of a linear transformer block: benefits of the mlp component and one-step gd initialization." NeurIPS.
>
> [2] Zhang, et al. "Trained transformers learn linear model in-context." JMLR.
>
> [3] Zhang, et al. "Transformer learns optimal variable selection in group-sparse classification." ICLR.
>
> [4] Frei and Gal. "Trained transformer classifiers generalize and exhibit benign overfitting in-context." ICLR.
>
> [5] Du, et al. "Algorithmic regularization in learning deep homogeneous models: layers are automatically balanced." NeurIPS.

---

> > ### Comment · Reviewer_enhE · 2025-04-05
> >
> > Thank you for your response, which addresses my main concerns. I will maintain my original score.

---

### Official Review · Reviewer_Rqgr · 2025-03-14

**Overall Recommendation:** 3

**Summary:**

This paper investigates the context hijacking phenomenon of transformer models, where incorporating multiple hijacking context samples can successfully flip the original model prediction. The paper conducts theoretical analysis on the linear transformer case for in-context learning, and verifies it on linear transformers/GPT-style transformers using a synthetic linear classification task. The experiments confirmed the theoratical analysis.

**Claims And Evidence:**

**Claim 1:**

Less hijacking in-context example and more transformer layers improve the transformer robustness against hijacking attack.

*Evidence:* The experiments in Section 5 aim to support this claim.

*Comment*: The experiment can confirm the theoretical analysis, but may be a bit limited since it mainly focuses on linear classification.

**Essential References Not Discussed:**

-

**Experimental Designs Or Analyses:**

Please see claims/evidence section.

**Methods And Evaluation Criteria:**

This work follows other work in theoretical analysis of transformer in-context learning. I think the evaluation is reliable.

**Other Comments Or Suggestions:**

I'm not very familiar with this domain and thus I'm not able to comment on the correctness of the proof. But the overall proof structure is clear and makes sense to me.

**Other Strengths And Weaknesses:**

-

**Questions For Authors:**

1. The paper is mainly motivated by the context hijacking, where the context itself is actually closely related to testing question/answer but not really the same. Are the distributions $D{te}$ and $D{tr}$ the same? If not, how does the difference between the two distributions affect the theoretical analysis?

**Relation To Broader Scientific Literature:**

This work is related to the theoretical analysis of transformer in-context learning works.

**Theoretical Claims:**

**Claim 1:**

Context hi-jacking phenomenon can be formulated following  previous modeling on transformer in-context learning.

*Evidence:* Section 3 formulates the problem.

*Comment:* The formulation mainly follows previous works on in-context learning analysis, and this paper extends the prior of $w^*$ to have non-zero mean, which better models testing a pre-trained model. Overall, it is reasonable to me.

**Claim 2:**

Testing error (context hijacking) can be formulated as a function of context length and number of layers in the transformer.

*Evidence:* Section 4.3 provides the proof.

*Comment:* I am not able to fully follow the analysis, but the proof structure is clean to me.

---

> ### Author Rebuttal · Authors · 2025-04-01
>
> Thank you for your recognition of our work and your constructive questions!
>
> >**Q1:**
> Are the distributions $\mathcal{D}\_{\mathrm{te}}$ and $\mathcal{D}\_{\mathrm{tr}}$ the same? How does the difference between the two distributions affect the theoretical analysis?
>
> **A1:**
> The distributions $\mathcal{D}\_{\mathrm{te}}$ and $\mathcal{D}\_{\mathrm{tr}}$ are not the same. $\mathcal{D}\_{\mathrm{tr}}$ is the distribution of sample $\mathbf{Z}$ during the training phase, and $\mathcal{D}\_{\mathrm{te}}$ is the distribution of sample $\mathbf{Z}\_{\mathrm{hc}}$ during the test phase (Section 3.1 and 3.3).
>
> Specifically, $\mathcal{D}\_{\mathrm{tr}}$ is a general in-context learning (ICL) data distribution modeling the pre-training distribution of a large language model. Consistent with common practices in theoretical studies on ICL [1-4], we consider classification pairs with Gaussian features.
>
> In contrast, $\mathcal{D}\_{\mathrm{te}}$ is a carefully designed data distribution that simulates the context hijacking phenomenon. We provide more intuitive explanations of its design in the following.
>
> For each sample $\mathbf{Z}\_{\mathrm{hc}}\sim \mathcal{D}\_{\mathrm{te}}$, its first $N$ columns are identical, consisting of $N$ repetitions of $(\mathbf{x}\_{\mathrm{hc}},y\_{\mathrm{hc}})$. Here, the $N$ repetitions of $(\mathbf{x}\_{\mathrm{hc}},y\_{\mathrm{hc}})$ represent the multiple repetitions of "Rafael Nadal is not good at playing" and "basketball" respectively in context hijacking example in Figure 1. The last pair $(\mathbf{x}\_{\mathrm{query}},y\_{\mathrm{query}})$ differs from $(\mathbf{x}\_{\mathrm{hc}},y\_{\mathrm{hc}})$, representing "Rafael Nadal’s best sport is" and "tennis". Additionally, we let the values of $\mathbf{x}\_{\mathrm{query}}$ and $\mathbf{x}\_{\mathrm{hc}}$ be closed, simulating the similarity between "Rafael Nadal is not good at playing" and "Rafael Nadal’s best sport is", while the corresponding $y\_{\mathrm{hc}}$ and $y\_{\mathrm{query}}$ have opposite signs, indicating the different answer "tennis" and "basketball".
>
> In summary, the context and query in $\mathcal{D}\_{\mathrm{tr}}$ are i.i.d., but the context in $\mathcal{D}\_{\mathrm{te}}$ are correlated to the query. As our key theory shows, multi-layer transformers perform multiple steps of gradient descent on the context samples. Therefore, when the distribution of the context is different, the gradient steps performed on the context will be significantly different.
>
> >**Q2:**
> The experiment may be a bit limited since it mainly focuses on linear classification.
>
> **A2:**
> We would like to clarify the motivation and organization of our paper. First we perform experiments on GPT2 using natural language data to identify the patterns of robustness of LLMs for context hijacking. Then we perform theory on linear classification, because linear problems have sufficient representation power, supported by many previous works [1-4]. Linear classification is a basic modeling of the problem. If we cannot effectively analyze linear problems, it is difficult for us to fully understand other problems. Based on linear problems, we build the first theoretical understanding of context hijacking and propose a comprehensive theoretical framework. We believe that based on our theoretical framework, people can expand to more complex classification problems (such as non-linear problems).
>
> We further verify our theoretical results, conducting experiments to validate our theory. Our experiments consider the optimal learning rate for gradient descent with different number of iterations, and the robustness for the linear transformer with different depths and the length of training context. The results are consistent with our theory. So our current experiments are developed to verify our theoretical analysis. And then we can bridge the gap between the theoretical results and the empirical findings on GPT2.
>
> Then, based on your suggestion, we further conduct some preliminary experiments on nonlinear classification (https://github.com/sfghtkgfv/dgnhjkgiqeb), changing $\langle \mathbf{w}, \mathbf{x}\rangle$ to $\langle \mathbf{w}, \mathbf{x}\rangle^2-C$, where $C$ is a constant. We conduct the experiment on multi-layer ReLU attention transformers. The results show that even in the nonlinear case, the model still tends to be more robust as it gets deeper, which is consistent with our theory.
>
> [1] Von Oswald, et al. Transformers learn in-context by gradient descent. ICML.
>
> [2] Ahn, et al. Transformers learn to implement preconditioned gradient descent for in-context learning. NeurIPS.
>
> [3] Zhang, et al. In-context learning of a linear transformer block: benefits of the mlp component and one-step gd initialization. NeurIPS.
>
> [4] Zhang, et al. Trained transformers learn linear models in-context. JMLR.

---

### Official Review · Reviewer_3ZRr · 2025-03-21

**Overall Recommendation:** 3

**Summary:**

The authors here have studied how the concept of context hijacking affects the transformer models. The context hijacking problem deals with the problem where giving some other informations to the model might affect it's output even if the informations are factually correct. The authors here tried to study this problem from both theoretical and practical aspects.

**Claims And Evidence:**

They claimed to have proved that deeper models help alleviate the context hijacking problem. I think the experimental evidence provided by them is convincing enough.

**Essential References Not Discussed:**

I don't think so

**Experimental Designs Or Analyses:**

They have used only linear transformers for testing their theory. I feel they could've given more clarity on it and why they chose the linear model.

**Methods And Evaluation Criteria:**

I think the evaluation method is fine. Though more clarity on the type of data they used and why would have been better.

**Other Comments Or Suggestions:**

I think there is lack of clarity or grammatical error in line 384, 'optimal gradient descent with more L steps'.

**Other Strengths And Weaknesses:**

The authors did very well in describing what context hijacking is. But they mentioned some terminologies like L-step gradient descent and L-transformers in the introduction which seemed unnecessary and confusing.

**Questions For Authors:**

No questions

**Relation To Broader Scientific Literature:**

I think the problem and their finding that increasing layers will help alleviate the context hijacking problem is quite general. Context hijacking seems like a general underfitting problem and increase in model complexity will help, this is a quite general result in my opinion.

**Theoretical Claims:**

I think more clarity on how they linked the problem with multi step optimization is needed.

---

> ### Author Rebuttal · Authors · 2025-04-01
>
> Thanks for your informative feedback! We address your comments as follows:
>
> >**Q1:** More clarity on the data model.
>
> **A1:**
> We model the context hijacking problem as a binary linear classification task, following general theoretical studies on transformer in-context learning [1, 2, 4, 5, 7]. Our context hijacking data consists of $\mathbf{x}\_{\mathrm{hc}}, y\_{\mathrm{hc}}, \mathbf{x}\_{\mathrm{query}}$ and $y\_{\mathrm{query}}$. Taking the left picture in Figure 1 as an example, we can assume that $\mathbf{x}\_{\mathrm{hc}}$ = “Rafael Nadal is not good at playing”, $y\_{\mathrm{hc}}$ = “basketball”, $ \mathbf{x}\_{\mathrm{query}}$ = "Rafael Nadal’s best sport is", and $y\_{\mathrm{query}}$ = "tennis".
>
> So our data structure is defined as follows. It consists of two parts: context and query. The first $n$ columns of our data represent context samples, where each column is a query-answer pair $(\mathbf{x}\_{\mathrm{hc}}, y\_{\mathrm{hc}})$. The last column of the sample contains a query $ \mathbf{x}\_{\mathrm{query}}$. This corresponds to the repeated context sentence "Rafael Nadal is not good at playing basketball" and the final query “Rafael Nadal’s best sport is” in practical case.
>
> >**Q2:** How to link the problem with multi-step optimization and why you chose the linear transformers? Additionally, unclear terms like $L$-step gradient descent and $L$-transformers, and a grammatical error.
>
> **A2:**
> Based on the classification task above, this paper aims to establish a rigorous theoretical analysis of the transformers' robustness against context hijacking, focusing on parameters such as depth $L$, context length $n$ an $N$, and embedding dimension $d$.
>
> However, existing transformer optimization analyses primarily focus on single-layer models [4, 5, 6, 7]. Analyzing the training processes of multi-layer transformers appears nearly impossible. Fortunately, recent works [1, 2, 3] provide a solid analytical framework for in-context learning of multi-layer linear transformers. Specifically, they demonstrated that **when given the in-context input matrix $\mathbf{Z}$ (eq. (3.1)), the corresponding output of $L$-layer linear transformers $\hat{y}\_{\mathrm{query}}$ (eq. (3.4)), is equivalent to that of a linear model trained via $L$-step gradient descent on all in-context pairs.** We adopt this framework and extend it in Lemma 4.1 to allow gradient descent to be initialized arbitrarily, which, as reviewer Rqgr noted, is a more reasonable conclusion.
>
> In summary, we consider linear transformers due to their proven in-context learnability, enabling us to derive derive clear insights into the robustness of multi-layer transformers (Theorem 4.5). Notably, a recent work [8] also employs linear transformers to study robustness against adversarial context in theory, but only considers one-layer models. Additionally, we conduct experiments on softmax attention transformers with GPT-2 style architectures, and the results (Appendix H.1) support our conclusions.
>
> Furthermore, we appreciate your feedback on unclear terms and grammatical errors; we will address them in revision.
>
> >**Q3:**
> Increasing the model complexity will mitigate context hijacking is a general result, as it appears to be an underfitting problem.
>
> **A3:**
> First, we would like to clarify that the focus of this paper, robustness against hijacking, differs from a fitting problem associated with the training data. This distinction arises as our test data is specifically designed to simulate the phenomenon of context hijacking, following a distribution that differs from that of the training data. Consequently, the overfitting or underfitting of the training data may not be directly connected to robustness.
>
> Figure 4 (Section 5.2) shows that shallow models experience underfitting issues. However, once the model depth exceeds 3, it can achieve an accuracy $\ge 99\\%$, indicating that transformers with 4 or more layers do not suffer from underfitting. In contrast, Figure 3 (Section 5.2) demonstrates that even when the model depth exceeds 3, the robustness of the model continues to improve as the depth increases.
>
> [1] Von Oswald, et al. Transformers learn in-context by gradient descent. ICML.
>
> [2] Ahn, et al. Transformers learn to implement preconditioned gradient descent for in-context learning. NeurIPS.
>
> [3] Bai, et al. Transformers as statisticians: Provable in-context learning with in-context algorithm selection. NeurIPS.
>
> [4] Zhang, et al. In-context learning of a linear transformer block: benefits of the mlp component and one-step gd initialization. NeurIPS.
>
> [5] Zhang, et al. Trained transformers learn linear model in-context. JMLR.
>
> [6] Zhang, et al. Transformer learns optimal variable selection in group-sparse classification. ICLR.
>
> [7] Frei and Gal. Trained transformer classifiers generalize and exhibit benign overfitting in-context. ICLR.
>
> [8] Anwar, et al. Adversarial robustness of in-context learning in transformers for linear regression. arXiv.

---

### Decision · Program_Chairs · 2025-05-01

**Decision:**

Reject

**Comment:**

Reviewers agree that the paper provides clear exposition and the main results may inspire follow-up works. However, reviewers have also found that a crucial ingredient of this paper is to translating linear transformer models to optimizing via gradient, which was established in prior works. This diminishes the technical contribution of this paper.